# Cannabinoid non-cannabidiol site modulation of TRPV2 structure and function

Liying Zhang[1,2,3,11], Charlotte Simonsen[1,11], Lucie Zimova [4], Kaituo Wang [3], Lavanya Moparthi [5,6], Rachelle Gaudet [7], Maria Ekoff[8], Gunnar Nilsson [8], Ute A. Hellmich [9,10], Viktorie Vlachova [4], Pontus Gourdon [2,3] ✉ & Peter M. Zygmunt [1] ✉

TRPV2 is a ligand-operated temperature sensor with poorly defined pharmacology. Here, we combine calcium imaging and patch-clamp electrophysiology with cryo-electron microscopy (cryo-EM) to explore how TRPV2 activity is modulated by the phytocannabinoid Δ⁹-tetrahydrocannabiorcol (C16) and by probenecid. C16 and probenecid act in concert to stimulate TRPV2 responses including histamine release from rat and human mast cells. Each ligand causes distinct conformational changes in TRPV2 as revealed by cryo-EM. Although the binding for probenecid remains elusive, C16 associates within the vanilloid pocket. As such, the C16 binding location is distinct from that of cannabidiol, partially overlapping with the binding site of the TRPV2 inhibitor piperlongumine. Taken together, we discover a new cannabinoid binding site in TRPV2 that is under the influence of allosteric control by probenecid. This molecular insight into ligand modulation enhances our understanding of TRPV2 in normal and pathophysiology.

The mammalian heat-activated transient receptor potential (TRP) vanilloid 2 channel (TRPV2) was identified shortly after the seminal discovery of TRPV1 in the mammalian sensory nervous system as a receptor activated by noxious heat and capsaicin[1,2]. TRPV2 shares approximately 50% sequence identity with TRPV1 and is expressed in heat-activated nociceptive primary afferents[1]. Both TRPV1- and TRPV2-expressing C-fiber primary afferents are lost in human skin from individuals with Norrbottnian congenital insensitivity to pain[3]. However, the study of TRPV2-deficient mice did not provide any evidence of TRPV2 being a thermosensor under normal physiological conditions or its involvement in heat hyperalgesia[4]. A physiological role of TRPV2 as a thermosensor may also be in doubt because of its high-temperature activation

threshold (>50 °C) and loss of heat sensitivity after pre-exposure to temperatures above its temperature activation threshold[1,5,6]. Rather, TRPV2 may have a physiological role as a redox- and ligand-operated channel[7,8] in a diverse population of spinal cord neurons as well as in immune cells involved in both adaptive and innate immunity[9–15].

The pharmacological profile of TRPV2 is less clear compared to TRPV1 and other chemosensitive thermosensors like TRPV3, TRPV4, TRPM8 and TRPA1. Indeed, the lack of selective TRPV2 agonists and antagonists has hampered investigations aimed at understanding the channel's function in normal and pathophysiology. TRPV2 can be activated by probenecid, 2-aminoethoxydiphenyl borate (2-APB) and cannabinoids, all of which are non-selective TRP channel

[1]Department of Clinical Sciences Malmö, Lund University, Malmö, Sweden. [2]Department of Experimental Medical Science, Lund University, Lund, Sweden. [3]Department of Biomedical Sciences, University of Copenhagen, Copenhagen, Denmark. [4]Department of Cellular Neurophysiology, Institute of Physiology, Czech Academy of Sciences, Prague, Czech Republic. [5]Wallenberg Centre for Molecular Medicine, Linköping University, Linköping, Sweden. [6]Department of Biomedical and Clinical Sciences, Linköping University, Linköping, Sweden. [7]Department of Molecular and Cellular Biology, Harvard University, Cambridge, MA, USA. [8]Division Immunology and Allergy Unit, Department of Medicine Solna, Karolinska Institutet, Karolinska University Hospital, Solna, Sweden. [9]Faculty of Chemistry and Earth Sciences, Institute of Organic Chemistry and Macromolecular Chemistry and Cluster of Excellence "Balance of the Microverse", Friedrich Schiller University Jena, Jena, Germany. [10]Center for Biomolecular Magnetic Resonance, Goethe-University, Frankfurt/Main, Germany. [11]These authors contributed equally: Liying Zhang and Charlotte Simonsen. ✉e-mail: pontus.gourdon@med.lu.se; peter.zygmunt@med.lu.se

modulators[16–21]. Nonetheless, studies how these ligands modulate the structure and activity of TRPV2 are needed to obtain a molecular insight into structural requirements for ligand interaction with TRPV2[19]. The first high-resolution structures of TRPV1[22] using cryo-EM paved the way for understanding how polymodal TRP channels are gated at a near atom level, and many cryo-EM structures of TRPV channels, including TRPV2, are now available[21,23–25]. Importantly, recent cryo-EM studies of TRPV2 have proposed ligand binding sites for both activators such as cannabidiol (CBD), resiniferatoxin (RTX), and 2-APB as well as the inhibitor piperlongumine (PLG)[26–30].

Several TRP channels (TRPV1-TRPV4, TRPA1 and TRPM8) are modulated by cannabinoids and may therefore be termed ionotropic cannabinoid receptors, suggesting a conserved cannabinoid binding site or mechanism linked to gating of these TRP channels[20,31]. Several cannabinoids have been shown to modulate TRPV2 activity. Among these, the phytocannabinoids $\Delta^9$-tetrahydrocannabinol ($\Delta^9$-THC) and CBD are the most effective agonists[20]. However, structural evidence of a direct interaction between TRPV2 and cannabinoids has so far only been demonstrated for CBD[27]. Due to its different chemical structure, it is possible that $\Delta^9$-THC targets other or additional cannabinoid binding sites within TRPV2. $\Delta^9$-THC mediates psychotropic effects by acting on the G protein-coupled cannabinoid receptor 1 (CB1). Looking for non-psychotropic cannabinoids, we previously identified $\Delta^9$-tetra-hydrocannabiorcol (C16), which only differs from $\Delta^9$-THC by having a truncated alkyl sidechain (methyl vs pentyl in $\Delta^9$-THC), as an anti-nociceptive acting on TRPA1 at the spinal cord level[32–34]. Importantly, this compound does not display agonistic activity on either the CB1 or the CB2 receptor[32].

Here, we show that C16 activates rat TRPV2 and that this activation is strikingly potentiated after pre-exposure to probenecid. This suggests that TRPV2 harbors distinct binding sites for C16 and probenecid that can act synergistically on TRPV2. Cryo-EM studies of rat TRPV2 confirmed that C16 and probenecid have different structural effects on the channel pore. Importantly, the synergistic effect of C16 and probenecid is also observed in HEK293 cells expressing mouse TRPV2 or human TRPV2, as well as in rat and human mast cells with endogenous protein levels. Therefore, our results can guide the development of more selective and powerful activators or inhibitors of TRPV2 that can be used as pharmacological tools to explore the physiological role of the TRPV2 channel in sensory neurons, and inflammatory and immune cells.

## Results and Discussion

### Ligand activation of heterologously expressed TRPV2

The phytocannabinoid C16 is a non-psychotropic $\Delta^9$-THC analogue with antinociceptive effects through interaction with TRPA1[32]. Here, we investigated its effect on TRPV2, since this channel is also activated by cannabinoids such as $\Delta^9$-THC and CBD[20]. C16 increased the intracellular calcium levels ($[Ca^{2+}]_i$) in HEK293T cells expressing either rat, mouse or human TRPV2 (Fig. 1). Higher C16 concentrations were needed compared to the activation of rat, mouse and human TRPA1 ($EC_{50}$ values 10–30 μM)[32,35], suggesting a lower sensitivity of the TRPV2 channels. While we could not test concentrations higher than 100 μM of C16 due to solubility issues, C16 could also activate human TRPV1 (Supplementary Fig. 1) with a potency similar to that observed for human TRPA1[32,35]. This is in line with observations that the cannabinoid and vanilloid ligand binding sites overlap in certain TRP channels[20,31,36].

Probenecid, an established TRPV2[17,37–39] and TRPA1 agonist[18], activates all TRPV2 orthologs in a concentration-dependent manner, although with lower sensitivity for human TRPV2 (Supplementary Fig. 2). Here, we also add TRPV1 as a target for probenecid, which is as potent and effective at activating human TRPV1 as human TRPA1 (Supplementary Fig. 3), further refuting the claim that probenecid is selective for TRPV2[17]. After observing a synergistic effect between C16 and probenecid in TRPV2 activation in preliminary experiments, we investigated this synergy further. Based on the concentration-dependent effects of probenecid alone (Supplementary Fig. 2), we chose a threshold concentration of probenecid of 1 mM for all TRPV2 orthologs. Preincubation with 1 mM probenecid significantly increased the $[Ca^{2+}]_i$ induced by C16 compared to the level induced in vehicle-treated rat, mouse and human TRPV2-expressing HEK293T cells (Fig. 1).

To further characterize the effects of C16 and probenecid on the activity-dependent properties of rat TRPV2, we measured whole-cell currents using either voltage ramps from −100 mV to +100 mV (0.4 V/s) delivered every 1 s from a holding potential of 0 mV or at steady-state with a holding potential of −60 or −70 mV, of which the latter holding potential was used in heat ramp experiments. C16 alone at the highest concentration tested (100 μM) produced small currents (−4.9 ± 1.3 and 10.5 ± 2.7 pA/pF at −100 and +100 mV; n = 10) of similar amplitude with repeated applications at 25 °C (Supplementary Fig. 4a). Although C16-evoked currents increased 2-fold at 42 °C, currents did not change much in amplitude with repeated applications. This suggests lack of C16 use-dependent effects at 25 °C and 42 °C under our experimental conditions (Supplementary Fig. 4a). To improve TRPV2 responses for studies of C16 concentration-dependent effects, we tried heat stimulation prior to C16 exposure leading to a dramatic sensitization of TRPV2 responses, a phenomenon also described for 2-APB[6]. However, the postheat responses induced by even 100 μM C16 at 25 °C were slow and far from reaching saturation within minutes (maximal rising slope of 80 ± 30 pA/s at both potentials; n = 3), rendering it difficult to determine C16 concentration-response curves (Supplementary Fig. 4b). Therefore, we used a protocol in which a temperature ramp from 25 °C to ~60 °C (5 °C/s) was first applied in the control extracellular solution at a holding potential of −70 mV, and then the currents evoked by voltage ramps were measured at a constant temperature of 42 °C. Under these optimized conditions, C16 was applied sequentially at concentrations of 10, 30, 60 and 100 μM, which allowed to estimate the $EC_{50}$ values and the cooperativity index for C16 activation at both membrane potentials (Supplementary Fig. 4c, d). The $EC_{50}$ value is similar to that in calcium imaging experiments (Fig. 1). At 25 °C, 3 mM probenecid alone did not activate TRPV2 at either negative or positive membrane potentials (Supplementary Fig. 5a). When the solution temperature was raised above 40 °C, probenecid elicited substantial current responses (−45.7 ± 12.3 pA/pF at −100 mV and 95.9 ± 24.1 pA/pF at +100 mV, 42 °C; n = 6) that did not increase with repeated applications (Fig. 2a, b, Supplementary Fig. 5a). Likewise, at a holding potential of −60 mV, a second exposure to probenecid (10 mM) at 25 °C did not produce a larger current (Fig. 2c, d). This together with the findings that concentration-response curves to probenecid performed twice in succession were identical (Supplementary Fig. 5c), suggests a lack of TRPV2 use dependence in response to probenecid, which is in contrast to the use-dependent effect of 2-APB on rat TRPV2[6]. Notably, the effect of probenecid was transient in whole-cell recordings when $Ca^{2+}$ was present in the extracellular solution (Fig. 2c and Fig. 3c). Likewise, in a study by Bang et al.[17], probenecid but not 2-APB evoked a transient current response in whole-cell patch-clamp recordings with $Ca^{2+}$ in the extracellular bath solution. Together, this could indicate that the transient response to probenecid is dependent on extracellular $Ca^{2+}$ and involves an activation mechanism distinct from that of 2-APB. However, transient responses are also observed for 2-APB and CBD at higher concentrations, visible just before washout, in $Ca^{2+}$-free recording solutions[29]. Perhaps, the strength and duration of the stimulus also contribute to the transient responses, reflecting different conformational states of the channel. It may well be that probenecid and other agonists at low concentrations are modulators and at higher concentrations are activators of TRPV2, leading to desensitization of TRPV2-mediated cellular responses that could be beneficial in a physiological relevant context as discussed for some TRPA1 modulators/activators[40,41].

Next, we studied the synergistic effect of C16 and probenecid on rat TRPV2 heterologously expressed in HEK293T cells using the optimized whole-cell patch-clamp configuration and experimental conditions as described above. Here, preincubation of the cells with either

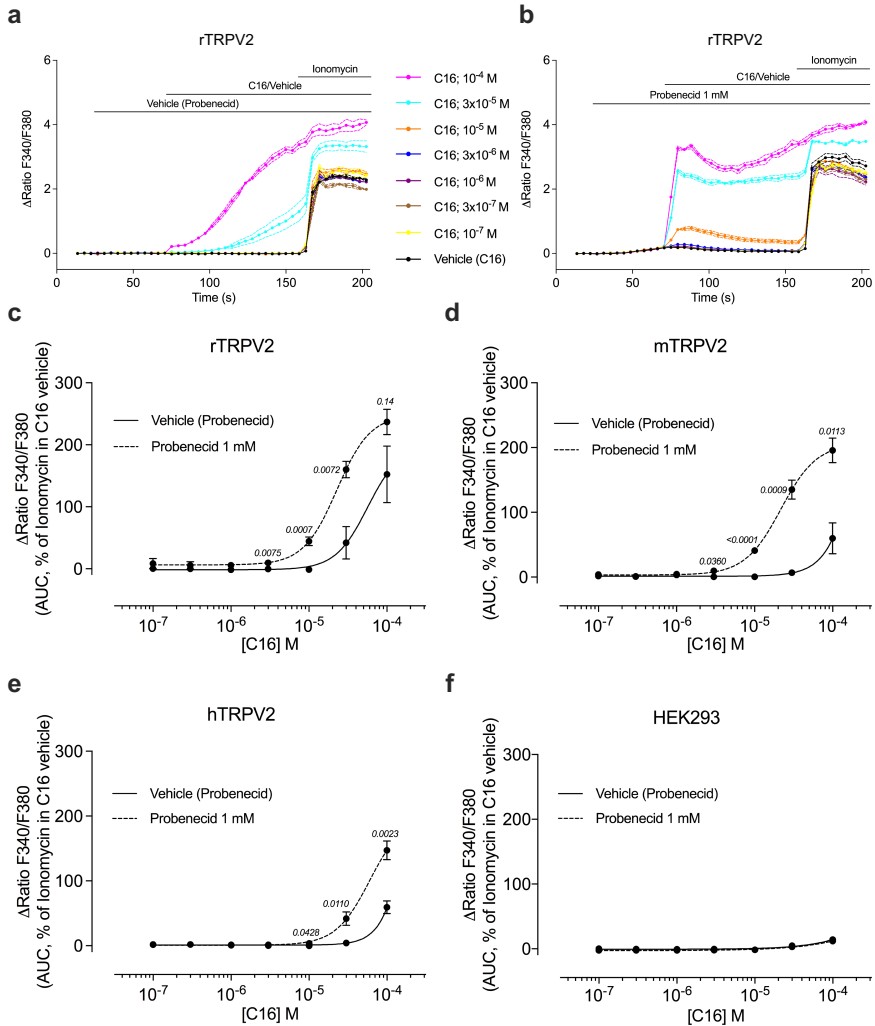

**Fig. 1 | C16 evoked TRPV2 responses are amplified by probenecid.** Probenecid (Prob, 1 mM), but not its vehicle, sensitized rat (**a-c**), mouse (**d**) and human (**e**) TRPV2-expressing HEK293T cells to the cannabinoid $\Delta^9$-tetrahydrocannabiorcol (C16) but not its vehicle. No amplification of C16 responses was observed in untransfected HEK293T cells (**f**). Data are shown as means ± SEM from three (**d**) to four (**c**, **e** and **f**) separate experiments, each performed in triplicates. The experiments were performed as shown in **a** and **b** with shared legends, where ionomycin (1 µM) was applied at the end of the experiment as positive control. The $[Ca^{2+}]_i$ responses were analyzed as area under the curve (AUC) for C16 in the presence of probenecid or its vehicle, and calculated as a percentage of the ionomycin response (AUC) in the presence of the C16 vehicle. P values are given in italics and indicate statistical significance when <0.05; Student's two-sided unpaired t-test. Source data are provided as a Source Data file.

probenecid or C16 followed by co-exposure with the other compound resulted in a significantly larger current than elicited by each compound alone (Fig. 2). This synergistic effect was inhibited by the recently proposed selective TRPV2 antagonist valdecoxib[39] at 100 µM (inhibition by 75.1 ± 1.9 % at −100 mV and 77.7 ± 4.3 % at +100 mV, n = 3) as well as by the nonselective TRP channel antagonist ruthenium red at 10 µM (Fig. 2e, f). Although ruthenium red at 10 µM can block TRPV2, a ten times higher concentration is usually needed to secure complete block of TRPV2[1,11,42,43]. We also studied the synergistic effect of probenecid and C16 using heat ramps from 25 °C to ~60–62 °C. Heat ramps applied in the extracellular control solution evoked inward currents in TRPV2-transfected HEK293T cells held at −70 mV (16 ± 3 pA/pF, n = 16) with an activation threshold of approximately 55 °C (Supplementary Fig. 6), which is in agreement with other studies[5,6]. Temperature-dependent currents were strongly sensitized by 3 mM probenecid and subsequent application of probenecid together with 10 µM C16 further enhanced activation (Supplementary Fig. 6b–d). Interestingly, the amplitude of the heat-induced current after wash-out of probenecid and C16 was similar to the initial heat response in the absence of these compounds, which is in line with findings that rat

TRPV2 heat responses after washout of 2-APB was similar in amplitude to that before application of 2-APB[5]. Furthermore, prior exposure to 2-APB did not sensitize heat activation but vice versa, suggesting separate activation pathways for ligands and heat[6]. Thus, in our heat ramp experiments it is likely that heat is sensitizing the effect of probenecid and C16, not the opposite, which is further substantiated by our experiments in which elevated temperature and a single prior heat ramp were used to improve ligand activation of TRPV2 (see e.g., Supplementary Figs 4 and 5). Furthermore, the heat response in the presence of probenecid after washout of C16 returned back to a magnitude similar to that before co-application of C16 (Supplementary Fig. 6b), and no effect of C16 at a subliminal concentration (10 µM), acting in synergy with probenecid, was observed in the absence of probenecid (Fig. 2a). Thus, both compounds need to be present for a synergistic effect on TRPV2. A synergistic activation of heterologously expressed TRPV2 has also been shown for 2-APB in combination with probenecid[39] or CBD[29] as well as 2-APB and oxidants[7]. Whether C16 acts synergistically with 2-APB, CBD or oxidants and how its gating mechanism is related to the heat activation pathway are topics for future studies.

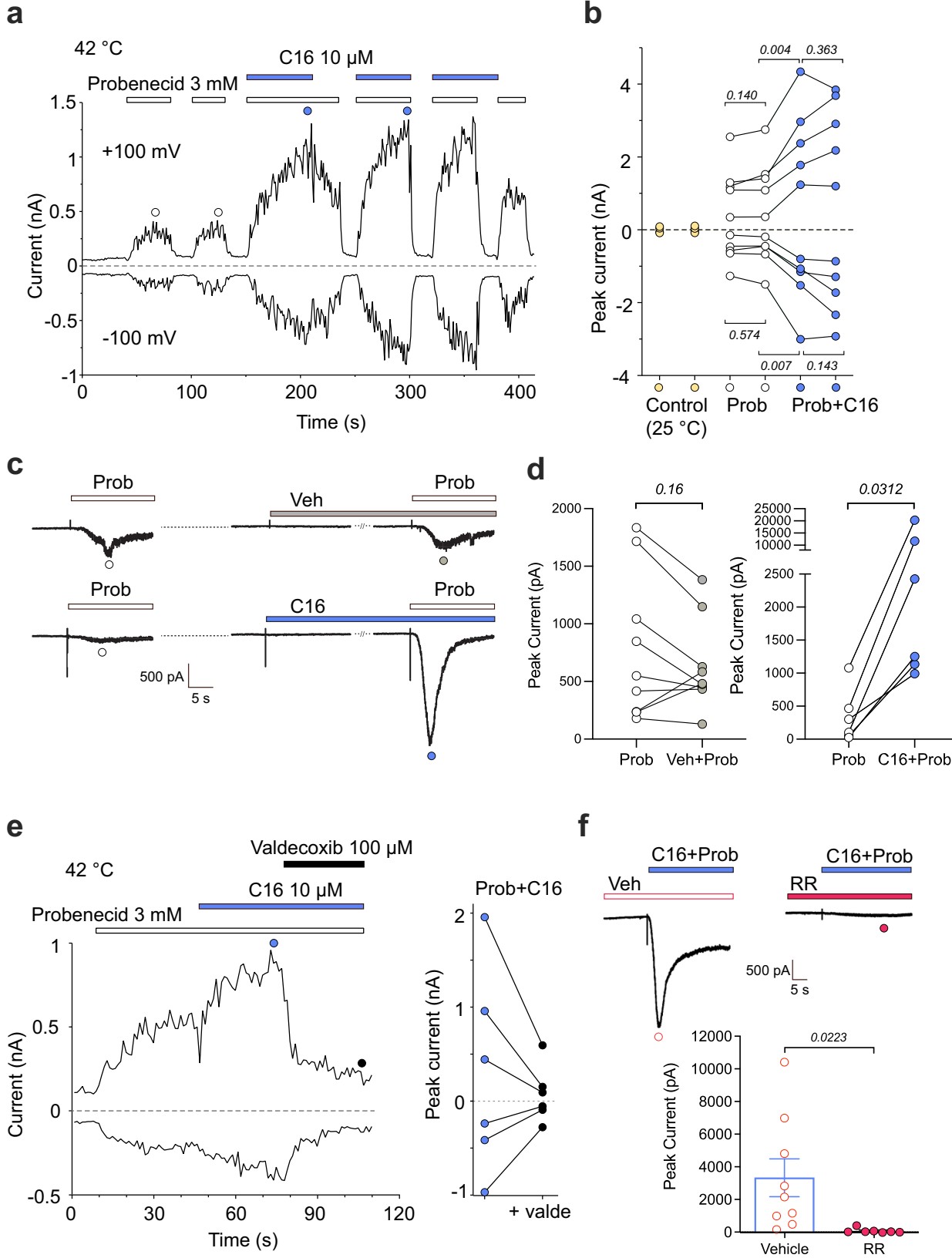

## Ligand activation of mast cells expressing TRPV2

The synergy of 2-APB and probenecid was also observed in the rat mast cell line RBL-2H3, expressing native TRPV2[9,11], although without reporting on physiological consequences such as mast cell degranulation and subsequent histamine release[39]. In addition to RBL-2H3,

both a human mast cell line (HMC-1.2) as well as human primary mast cells (CBMC) express TRPV2[9,11]. These cells were therefore used to study the effect of C16 and probenecid on native TRPV2. Individually, probenecid and C16 increased the $[Ca^{2+}]_i$ in RBL-2H3 cells only at the highest concentrations tested (Fig. 3a). As shown above for TRPV2-

**Fig. 2 | Synergistic effects of C16 and probenecid on rat TRPV2 currents in HEK293T cells.** Representative whole-cell current recording from HEK293T cell expressing rat TRPV2. **a** Inward and outward currents were elicited by voltage ramps from −100 mV to +100 mV (0.4 V/s) delivered every 1 s from a holding potential of 0 mV. Stimuli of 3 mM probenecid (white horizontal bars) and 10 μM C16 (blue horizontal bars) were individually or synergistically applied to the cell at 42 °C. **b** Statistical summary of the peak currents (open and blue circles) in (**a**), induced by probenecid (Prob) and probenecid with C16 (Prob + C16) at +100 mV and −100 mV. Control (yellow circle) indicates currents measured at 25 °C before increasing the temperature to 42 °C at which the effects of probenecid and C16 were recorded. Data are shown as means ± SEM from five separate experiments. **c** Representative continuous whole-cell inward current traces at a holding potential of −60 mV showing an initial exposure to probenecid alone (Prob, 10 mM) followed by a second application of probenecid (10 mM) after preincubation of either C16 at a subthreshold concentration of 10 μM (see Fig. 1c) or its vehicle (Veh). Each treatment is from a different cell. The co-application of probenecid and C16 only

induced a small current in control HEK293T cells (−39 ± 11 pA, $n = 4$ cells). **d** As summarized in the graph, the peak currents (open, grey and blue circles) in (**c**), evoked by probenecid increased in the presence of C16 ($n = 6$ independent experiments) but not in the presence of vehicle ($n = 9$ independent experiments). **e** The TRPV2 inhibitor valdecoxib (100 μM) inhibited outward and inward currents evoked by the combined application of probenecid and C16 (data are from three independent experiments performed as shown by the traces). **f** The non-selective TRP channel antagonist ruthenium red (RR, 10 μM, $n = 7$ independent experiments) inhibited currents evoked by the combined application of C16 (10 μM) and probenecid (10 mM) at a holding potential of −60 mV ($n = 9$ independent experiments). P values are given in italics and indicate statistical significance when <0.05, Student's two-sided paired t-test (**b**), Wilcoxon two-sided matched-pairs test (**d**) and Welch's two-sided unpaired t-test (**f**). Data are shown as means ± SEM. **a**, **b**, **e** Experiments recorded in extra- and intracellular Ca²⁺-free solutions. **c**, **d** and **f** Experiments recorded in extracellular Ca²⁺-containing solution. Source data are provided as a Source Data file.

expressing HEK293T cells, preincubation with 10 μM C16 potentiated the effect of 3 mM probenecid in RBL-2H3 cells. The synergistic effect of C16 and probenecid was reduced by 100 μM ruthenium red, but not by the TRPA1 and TRPV1 specific antagonists HC030031 (100 μM) and capsazepine (3 μM), respectively. Because RBL-2H3 cells are known to express both channels[9], we wanted to unambiguously rule out a contribution of TRPV1 and TRPA1 to the calcium influx observed in response to the synergistic stimulation by probenecid and C16. Thus, we also characterized the response of RBL-2H3 cells to probenecid in combination with the known TRPA1 and TRPV1 agonists mustard oil (100 μM) and capsaicin (1 μM), respectively (Fig. 3b). Neither agonist was able to potentiate the effect of probenecid (Fig. 3b).

We also observed synergy between probenecid and C16 in RBL-2H3 cells in whole-cell patch-clamp recordings, where probenecid (10 mM) induced a small current that was clearly potentiated by preincubation with 10 μM C16 (Fig. 3c). As with rat TRPV2 expressed heterologously in HEK293T cells, the response in RBL-2H3 cells was inhibited by ruthenium red (10 μM).

We further investigated the physiological consequences of TRPV2 activation in mast cells by studying histamine release. The synergy of TRPV2 activation resulted in an enhanced degranulation in RBL-2H3 cells (Fig. 3d). C16 alone did not cause histamine release in RBL-2H3 cells, while probenecid on its own triggered a significant release of histamine. Furthermore, the combination of C16 and probenecid significantly enhanced the histamine release compared to probenecid alone (Fig. 3d). The release of histamine was inhibited by 10 μM ruthenium red (Fig. 3d). The synergy of TRPV2 activation by the two compounds was even more prominent in human primary mast cells. Here, neither C16 nor probenecid had any effect alone, but combined evoked substantial release of histamine that was inhibited by 10 μM ruthenium red (Fig. 3d). C16 and probenecid, only when combined, also increased the [Ca²⁺]ᵢ in the human mast cell line HMC-1.2 (Fig. 3e). In contrast, neither mustard oil nor capsaicin increased [Ca²⁺]ᵢ in HMC-1.2 cells (Fig. 3e). Taken together, these findings encourage further analysis of how TRPV2 integrates and transforms various ligand interactions into physiological responses such as mast cell degranulation.

### Ligand activation of purified TRPV2

To confirm a direct interaction between C16 and TRPV2 before proceeding to cryo-EM studies, we performed experiments with purified rat TRPV2 reconstituted into artificial lipid bilayers for electrophysiological recordings with a focus on C16, since probenecid activation of purified rat TRPV2 reconstituted in lipid bilayers had already been demonstrated[37]. As shown in experiments measuring currents between −100 and +100 mV, purified rat TRPV2 responded to C16 (100 μM) with currents at both negative and positive test potentials (Supplementary Fig. 7a). At steady state, using holding potentials of

+60 or −60 mV (Supplementary Fig. 7b–d), distinct single-channel openings were observed that were inhibited by 40 μM ruthenium red (Supplementary Fig. 7c). The C16-evoked single-channel conductance and open probability values were 50 ± 3 pS ($n = 6$) and 0.45 ± 0.06 ($n = 4$), respectively, at +60 mV. The single-channel open probability at +60 mV was reduced by ruthenium red to 0.10 ± 0.05 ($n = 3$) ($P = 0.01$, Student's two-sided paired t-test). Bilayers without TRP channels do not respond to C16 (Supplementary Fig. 7b) as has also been shown previously[35]. We also confirmed that probenecid activates purified rat TRPV2 when reconstituted into lipid bilayers[37] ($n = 2$, Supplementary Fig. 7d).

### Cryo-EM structures of TRPV2 in the presence of C16

To date, the effects of RTX, CBD, PLG, and 2-APB on TRPV2 structure-function have been studied using cryo-EM or X-ray crystallography[27,28,30,44,45] (Supplementary Fig. 8a, b). Here, we present two distinct cryo-EM structures of rat TRPV2 in the presence of C16, TRPV2_C16-1 and TRPV2_C16-2, which were obtained at an overall resolution of 2.9 Å and 3.2 Å, respectively (Fig. 4). As expected from the previous studies[21,24,36], the channel architecture displays the classical domain-swapped homotetrameric assembly of TRP channels. Each subunit harbors a transmembrane domain of six transmembrane segments (S1–S6) with S1–S4 forming a separate voltage sensing-like domain, while S5-S6 and the intervening pore helix (PH) form the pore domain that lines the ion-conducting pathway. Moreover, a large soluble domain is present, consisting of six ankyrin repeats forming the ankyrin repeat domain (ARD) and two β-sheets in the N-terminus, as well as a C-terminal TRP-helix domain containing both a long and a short β-sheet (Supplementary Fig. 8a, b). As observed in previous TRPV2 structures, the resolution of these soluble components, especially ankyrin repeats 1 and 2, is lower than the transmembrane part[24]. These less defined regions are still sufficiently well resolved for modeling of the backbone, but not of the sidechains. The ion-conducting pathway is in the center of the tetramer and features a selectivity filter (SF) including the PH at the cytoplasmic membrane interface. At the intracellular end of the ion-conducting pathway, a lower gate (LG) is lined by S6. Together, the LG and SF represent the two classical key molecular determinants that orchestrate TRP channel-mediated ion flow[21,24,36].

### C16 effects on the ion-conducting pathway

Two structures of TRPV2 in the absence of ligands (apo) have been reported, displaying different pore width profiles along the ion-conducting pathway[27]. TRPV2_apo-1 (PDB-ID: 6U84) with equally constricted SF and LG, and TRPV2_apo-2 (PDB-ID: 6U86) with a relatively open SF configuration but a constricted LG as in TRPV2_apo-1[27]. The opening of SF in TRPV2_apo-2 was proposed to be dependent on intrinsic conformational plasticity of the loops connecting the PH and S6[44].

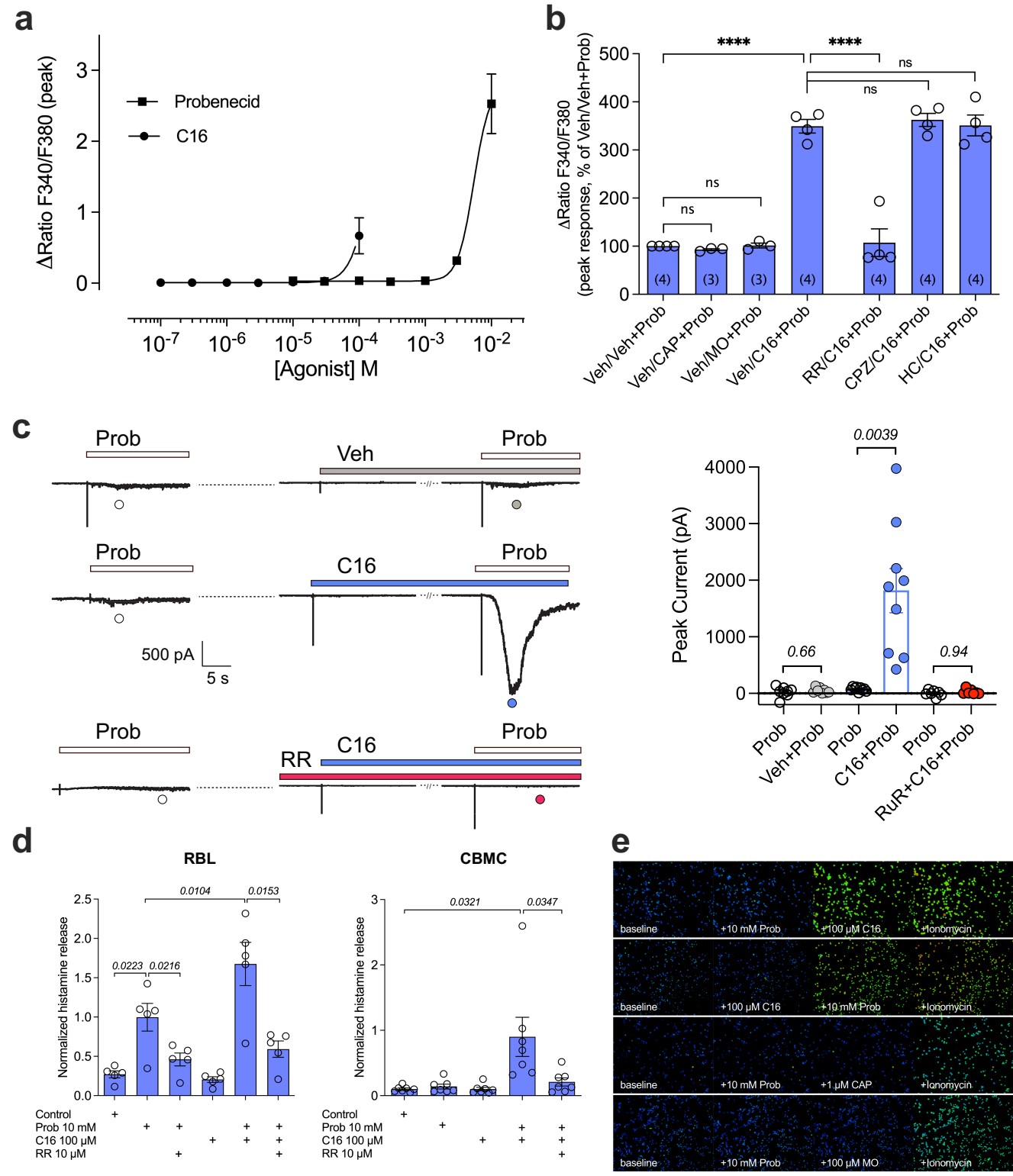

Conversely, the SF and LG of TRPV2 can widen simultaneously as revealed by an open TRPV2$_{open}$ structure (PDB-ID: 6BO4)[46]. We compared the pore region of TRPV2$_{apo-2}$, TRPV2$_{open}$, TRPV2$_{C16-1}$ and TRPV2$_{C16-2}$ by computing pore radii based on polyalanine models. This was done by mutating residues 504 to 651, along the ion conduction pathway, to alanine in silico to minimize the effect of differences in the resolution of the compared structures, as secondary structure elements are more accurately assigned at moderate resolutions relative to

sidechains. The advantage of this approach is that it accounts for the fact that sidechains are difficult to model at a resolution lower than 3 Å. This allows more reliable comparisons between the relative differences between structures, even though it of course needs to be considered that the generated absolute values do not represent the factual pore widths, since sidechains naturally also assist in forming the conducting pathways. As expected, the two structures obtained in the presence of C16 displayed relatively open configurations compared to TRPV2$_{apo-1}$,

**Fig. 3 | C16 and probenecid are synergistic activators of rat and human mast cells. a** Concentration-dependent elevation of $[Ca^{2+}]_i$ produced by individual probenecid and $\Delta^9$-tetrahydrocannabiorcol (C16) application in RBL-2H3 cells. C16 concentrations tested were limited because of its solubility limit. Data are shown as means ± SEM from three (C16) to four (probenecid) independent experiments performed in duplicate or triplicate. **b** Preincubation of RBL-2H3 cells with C16 (10 μM), but not capsaicin (CAP, 1 μM) or mustard oil (MO, 100 μM) potentiated the response to probenecid (3 mM). The $[Ca^{2+}]_i$ increase induced by C16 and probenecid co-treatment was inhibited by the non-selective TRP antagonist ruthenium red (RR; 100 μM) but not by the TRPV1 antagonist capsazepine (CPZ, 3 μM) or the TRPA1 antagonist HC030031 (HC, 100 μM); one-way ANOVA followed by Sidak's multiple comparisons test: $F_{6,19} = 64.86$; ****$P < 0.0001$ and ns = not significant ($P \geq 0.9940$). Data are shown as means ± SEM from three to four independent experiments, as indicated within parentheses, each performed in triplicate. **c** As shown by traces from continuous recordings in RBL-2H3 cells (each from a separate cell), preincubation with C16 (10 μM), but not its vehicle (Veh), potentiated the whole-cell current induced by a second application of probenecid (Prob, 10 mM), whereas ruthenium red (RR, 10 μM) prevented probenecid-induced currents. Experiments were recorded in extracellular $Ca^{2+}$-containing solution. As summarized in the bar graph, the average peak current induced by the second application of probenecid in the presence of C16 ($n = 9$), but not its vehicle ($n = 8$) or ruthenium red ($n = 7$) was significantly larger compared to the peak current induced by the first application of probenecid (P values are given in italics and indicate statistical significance when <0.05; Wilcoxon two-sided matched-pairs test). Data are from independent experiments and shown as means ± SEM. **d** The combination of probenecid and C16 more effectively triggered a ruthenium red sensitive histamine release from RBL-2H3 ($n = 5$ independent experiments) and human mast cells (CBMC, $n = 7$ biologically independent donors) than the application of the individual compounds. Data are shown as means ± SEM, and P values given in italics indicate statistical significance when <0.05; repeated measures one-way ANOVA followed by Sidak's multiple comparisons test: $F_{1.062,4.249} = 34.60$ (RBL) and Student's two-sided paired t-test (CBMC). **e** Probenecid combined with C16, but neither capsaicin (CAP) nor mustard oil (MO), increased intracellular calcium in the human mast cell line (HMC-1.2) expressing native TRPV2 ($n = 2$-3 independent experiments). The calcium ionophore ionomycin (1 μM) was used as positive control. Source data are provided as a Source Data file.

in agreement with a functional response to the ligand. However, surprisingly, while TRPV2$_{C16-1}$ appears enlarged in the SF region and constricted at the LG, TRPV2$_{C16-2}$ demonstrates an inverse pore profile, being wider at the LG (Fig. 4a). As such TRPV2$_{C16-1}$ has a pore profile similar to that of the previously determined TRPV2$_{apo-2}$ (Fig. 4b). Thus, TRPV2$_{C16-1}$ resembles an apo structure (TRPV2$_{apo-2}$), with the above-mentioned auto-stimulated open configuration of the SF and constricted LG. This is also consistent with root mean square deviation (RMSD) comparisons of the structures and subdomains (Supplementary Table 1). In contrast, our TRPV2$_{C16-2}$ structure with a wider LG may, in combination with spontaneous SF opening, reach a pore-open state. We conclude that the substantial pore profile differences between TRPV2$_{C16-2}$ and previously determined apo states most likely are caused by C16, and it cannot be excluded that C16 has affected the channel pore also in TRPV2$_{C16-1}$ (Fig. 4b).

## C16 effects on the vanilloid pocket region

The vanilloid pocket, located between the S1–S4 bundle of one subunit and the S5-S6 pore domain of an adjacent subunit[22], is a common feature of the TRPV subfamily that is also shared with several other TRP subfamilies[25,36]. Through binding to this cavity in TRPV1, ligands such as capsaicin, RTX and capsazepine cause activation and/or inactivation/desensitization of TRPV1[22,47–51]. However, non-vanilloids may also associate within this pocket and similar ligand pockets appear in other TRP channels pointing towards a more general and key ligand TRP modulatory site[25,36,52]. Notably, in the TRPV2$_{CBD}$ and TRPV2$_{apo}$ structures, the compartment is occupied by lipids[27]. However, ordered lipids were not detected in TRPV2$_{open}$[46], which hints that the binding of certain lipids have an inhibitory role on membranous ion passage of TRPV2[46] as originally proposed for TRPV1[22,47], although it cannot be excluded that a lipid simply could not be resolved in TRPV2$_{open}$.

Here, by comparing our cryo-EM maps with previously published TRPV2 structures[27], we observe non-protein cryo-EM density in the vanilloid pocket of TRPV2$_{C16-1}$ (Fig. 4c). Considering the overlapping shape and size of the cryo-EM density as well as the local chemical environment, we assign this density to a lipid with a phosphate headgroup such as phosphatidylcholine (PC) (Fig. 4c, e, & Supplementary Fig. 9a, Supplementary Fig. 10), similar to what has been observed at the corresponding site in structures of TRPV3[53,54]. However, we cannot exclude that the lipid is phosphatidylinositol (PI), as observed in TRPV1 structures[22,47,49,55], or another type of lipid. The phospholipid headgroup is positioned between S3, S4, and the TRP-helix, where its negative charge and polar groups are stabilized by positively charged R517 and K531, and the polar residue Q663 (Fig. 4e). Notably, the equivalent residues are known to stabilize lipids in this location in TRPV1 and TRPV6[47,56]. The acyl tails extend into the vanilloid pocket as

observed in TRPV1 and TRPV3[22,47,49,53,55]. We note that both PC and PI are prevalent in the expression host *Pichia pastoris*, congruent with co-purification of either PC or PI from the protein expression host[57]. In our analysis of currently available TRPV2$_{apo}$ structures and maps, for which the protein was produced in *Saccharomyces cerevisiae*[27], we found density consistent with two acyl tails in the same region of the vanilloid pocket (Supplementary Fig. 10).

We also observed non-protein cryo-EM density in the vanilloid pocket of TRPV2$_{C16-2}$ (Fig. 4d). However, this feature is smaller, and its shape differs from the density we assigned to phospholipids in TRPV2$_{C16-1}$, TRPV2$_{apo}$ and TRPV3 structures (Fig. 4f, Supplementary Fig. 10). Indeed, through comparison of TRPV2$_{C16-1}$ with TRPV2$_{apo-2}$, and TRPV2$_{C16-1}$ with TRPV2$_{C16-2}$, the cavity hosting the lipid headgroup in TRPV2$_{C16-1}$ appears shrunken in TRPV2$_{C16-2}$ (from ~280 to ~180 Å$^3$ as estimated using the software Chimera), (Fig. 4f, Supplementary Fig. 9b, Supplementary Fig. 11a–c). This indicates that the lipid of TRPV2$_{C16-1}$ has been replaced, and we have instead tentatively assigned a cholesteryl hemisuccinate (CHS) at this location in TRPV2$_{C16-2}$, based on the shape of the feature. CHS is included in the protein preparation and could mimic native cholesterol, and is also in agreement with the equivalent binding of cholesterol to a previous structure of TRPV2[30] (PDB-ID: 7XEM) and CHS to TRPV6[58] (PDB-ID: 7S89) (Supplementary Fig. 12). Notably, the CHS succinate head, located in the interface between S5-S6 of one subunit and the S4-S5 linker of an adjacent subunit, points towards same positively charged residues R517 and K531 as the PC headgroup (Fig. 4f, Supplementary Fig. 11a, b), although the sidechains have reorganized (Supplementary Fig. 9b, Fig. 4f Supplementary Fig. 11a, b).

Interestingly, the feature of the vanilloid pocket in TRPV2$_{C16-2}$ partially overlaps also with the RTX bound in structures of engineered TRPV2[44,59–61] (Supplementary Fig. 13). The sidechains of residues Y544, T604 and Y629 form a hydrogen-bond triad network that is disrupted by RTX binding, facilitating stabilization of the LG of TRPV2 in a semiopen state[61]. We make the same observation in TRPV2$_{C16-2}$. Thus, C16-induced modulation of the vanilloid pocket in TRPV2$_{C16-2}$ may have similar molecular consequences as the binding of RTX, leading to the widening of the LG upon binding of C16.

## Putative C16 binding site

In addition to the density assigned to CHS, another feature is present in the TRPV2$_{C16-2}$ map (Figs. 4d, f, 5a), present in both so-called half-maps, further suggesting it represents a ligand rather than noise (Supplementary Fig. 14). We tentatively assigned this density to C16 because of the associated structural changes in the channel pore region (Fig. 5a, b), and because of the lack of a similar density in other maps of available TRPV2 structures. However, the size and shape of

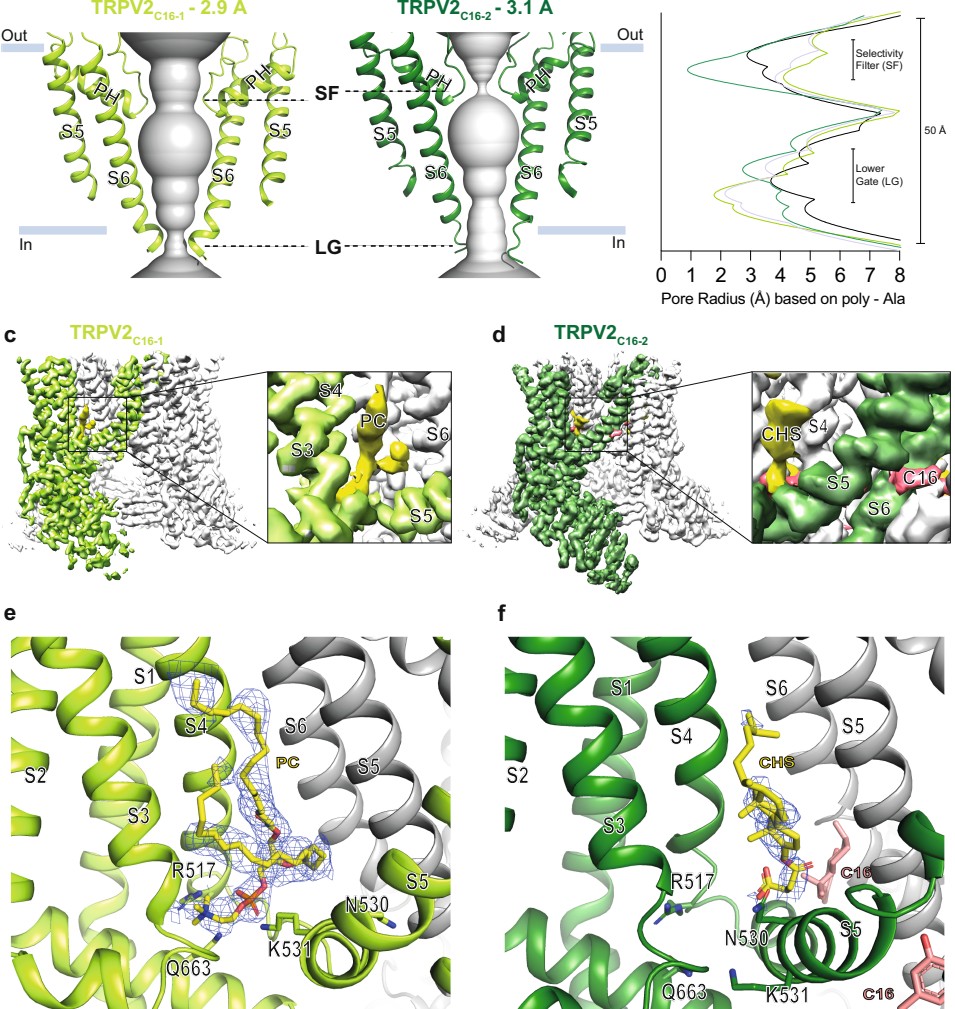

**Fig. 4 | Molecular effect of C16 and the vanilloid pocket. a, b** Profile of the ion permeation pathway, displayed as gray space-filling models, for (**a**) TRPV2$_{C16-1}$ (light green) and TRPV2$_{C16-2}$ (dark green) and (**b**) the corresponding calculated radii as a function of the distance along the ion conduction pathway. The pore radii of TRPV2$_{open}$ (PDB-ID: 6BO4) and TRPV2$_{apo-2}$ (PDB-ID: 6U86) are also included as references and shown by black and white-blue lines, respectively. All pore calculations in this and subsequent analyses are based on polyalanine exchange of residues 504 to 651 including the pore helix (PH), selectivity filter, and lower gate to

minimize the effect of differences in the resolution between structures. **c, d** Overall cryo-EM maps for (**c**) TRPV2$_{C16-1}$ (light green) at 2.9 Å and (**d**) TRPV2 $_{C16-2}$ (dark green) at 3.1 Å, with close-views of the vanilloid pocket showing putative (**c**) phosphocholine (PC, yellow), (**d**) CHS (yellow) and C16 (pink) features. **e, f** Details of the vanilloid binding pocket with (**e**) PC (yellow sticks) in TRPV2$_{C16-1}$ and (**f**) CSH (yellow sticks) in TRPV2$_{C16-2}$. C16 is shown as pink sticks in **f**. Key selected amino acids are shown as sticks. The corresponding cryo-EM density features of PC and CSH, but not C16, are shown as blue mesh (5.0 σ).

the density does not completely match C16, which may relate to the limited resolution and the C16 occupancy. Indeed, the density is also compatible with an alternative C16 pose in the binding site (Supplementary Fig. 14). The same region of the TRPV2$_{C16-1}$ map also shows density, distinct from PC, although it cannot be unambiguously assigned to C16 (Fig. 5c). Consistent with C16 binding, residues H521 and Y525 that line the C16 pocket rotate considerably between TRPV2$_{apo-1}$ and our two structures to provide space for the compound (Fig. 5a, d). As such, TRPV2$_{C16-1}$ may represent an intermediate state where the ligand has bound but the PC has not yet been displaced. Interestingly, the proposed binding site for PLG[28] is close to the C16 binding site, but without the H521 and Y525 movements (Fig. 5e). Notably, no indication of a ligand at the CBD site is present in our structures obtained from C16-treated sample. The putative C16 binding is at a different position than that proposed for CBD in TRPV2$_{CBD}$, which however is a closed structure and retains a lipid in the vanilloid pocket (Fig. 5a). Finally, an important observation from the comparisons of the structures determined is that C16 affects TRPV2 beyond the local binding pocket, and thereby significantly

increasing the pore width along the conduction pathway (Fig. 5f, and see below).

To test which of the proposed C16 binding configurations is most likely to trigger C16 activity, we examined the functional role of four critical residues (D536, F540, S526 and L636) around the proposed binding site using site-directed mutagenesis (Fig. 6; Supplementary Figs. 15 and 16). In cells expressing the D536E mutant, C16 elicited no response up to 100 μM (Fig. 6). This construct, on the other hand, exhibited significant currents in response to either probenecid or heat, although currents were reduced with repeated exposure to each stimulus, and a shift to higher temperatures for heat activation was observed (Supplementary Figs. 15 and 16). The temperature-dependent currents were strongly potentiated by probenecid, but desensitized upon repeated applications (Supplementary Fig. 15c). There was no synergistic effect with co-application of C16 either using heat ramps (Supplementary Fig. 15c–e) or at constant heat (50 °C) at negative and positive membrane potentials in voltage ramp experiments (Supplementary Fig. 15f). It is most likely that D536E because of its longer sidechain prevents the channel from interacting with C16.

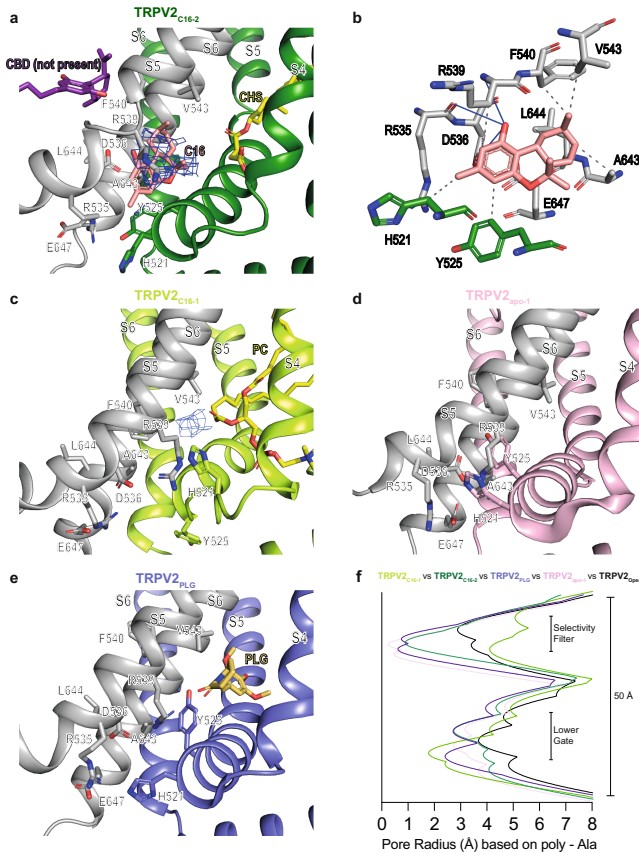

**Fig. 5 | C16 binding.** Proposed interaction site of C16 at the interface of two TRPV2 subunits. **a** TRPV2$_{C16-2}$ (dark green) and one adjacent subunit in grey with C16 shown as pink sticks and the corresponding cryo-EM density as blue mesh (3.0 σ). Shown is also the binding site for cannabidiol CBD, which is distinct from the C16 binding site (adapted from PDB-ID: 6U8A). **b** The interaction network with C16 (pink sticks) and nearby residues in TRPV2C$_{16-2}$. **c** The TRPV2$_{C16-1}$ structure (light green) also shows cryo-EM density (blue mesh at 3.0 σ level, at which protein sidechains are distinguishable) in the assigned C16 binding region of TRPV2C$_{16-2}$, although we have left it unmodeled. CHS (**a**) and PC (**c**) are shown as yellow sticks. **d**, **e** The corresponding site where we observe density for C16 in our structures (**a**, **c**) is shown in TRPV2$_{apo-1}$ (PDB-ID: 6U84, pink) and TRPV2$_{PLG}$ (PDB-ID: 6WKN, purple). PLG is shown as yellow sticks. **f** The pore radius in TRPV2 cryo-EM structures as a function of the distance along the ion conduction pathway. TRPV2$_{C16-1}$ (light green), TRPV2$_{C16-2}$ (dark green), TRPV2$_{apo-1}$ (PDB-ID: 6U84, pink) and TRPV2$_{PLG}$ (PDB-ID: 6WKN, purple). The pore radius of TRPV2$_{open}$ (PDB-ID: 6BO4) is included for reference (black).

Taken together, these data indicate that D536 is a crucial residue involved in the interaction of TRPV2 with C16, and of importance for a proper synergistic interaction between the C16 binding site and a separate binding site for probenecid. In contrast, the S526A mutant was fully functional, and its concentration dependence of C16-evoked responses was not much different from wild-type TRPV2 (Fig. 6b). Cells expressing the F540A mutant failed to respond with currents induced by heat, voltage, probenecid and C16 applied individually or synergistically why this mutant was not further studied here. The change in TRPV2 heat responsiveness for some mutants reflected by the significant increase in currents at 42 °C (S526A, S526E and L636A) and a higher temperature activation threshold (D536E and S526E) compared to wild type TRPV2 (Supplementary Fig. 16) may suggest that the C16 binding pocket is influenced by a separate heat pathway[6]. Nevertheless, the inhibitory effect of C16 as revealed by the S526E and L636A mutants (Fig. 6c, d), further indicates that targeting the C16 binding site may be a suitable approach to fine tune TRPV2 activity using agonists and possibly inverse agonists. Collectively, these data points

towards C16 being bound as shown in the first binding configuration (Fig. 5a, b).

## Analysis of C16-induced conformational changes in TRPV2

To evaluate the mechanistic consequences of the C16-TRPV2 interaction, we compared TRPV2$_{C16-2}$ with TRPV2$_{apo-1}$, and TRPV2$_{C16-1}$ with TRPV2$_{apo-2}$, respectively, to minimize the effect of the conformational differences of the SF (Supplementary Table 1). Structural alignments were performed using global comparisons of the tetrameric channel structures. The TRPV2$_{C16-2}$ and TRPV2$_{apo-1}$ structures differ within the pore domain (Fig. 7a). Specifically, the TRPV2$_{C16-2}$ S5 helix shifts towards the extracellular side and the preceding S4-S5 linker rotates to compensate the movement of S5, which in turn affects the S6 helix at the LG region via interactions with Y525 and E647 (Fig. 7b, c, Supplementary Fig. 17). Consequently, the loop connecting S6 and the TRP-helix (residues 651–655 in TRPV2$_{apo-1}$) extends to include residues 647–655 in TRPV2$_{C16-2}$, thereby widening the channel at the LG (Fig. 7c). The absence of lipid with a phosphate headgroup (such as PC) in TRPV2$_{C16-2}$ contributes to these changes, allowing space for shifts of S1, S4 and the TRP-helix towards the ion-conducting pore (Fig. 7b, c).

In contrast to the structural differences between TRPV2$_{C16-2}$ and TRPV2$_{apo-1}$, the differences between TRPV$_{C16-1}$ and TRPV2$_{apo-2}$ are more subtle and influence only specific channel regions. In particular, the intracellular end of S5 and the TRP-helix are affected (Supplementary Fig. 18). Specifically, in TRPV$_{C16-1}$, a remarkable change of Y525 sidechain is observed, flipping towards the cytosol (Fig. 5c), as in TRPV2$_{C16-2}$ (Fig. 5a). However, the LG remains closed in TRPV2$_{C16-1}$ compared to TRPV2$_{C16-2}$, perhaps due the presence of lipid in the vanilloid pocket (Supplementary Fig. 18b, Fig. 4c, e). Based on the two C16-bound structures, we propose a lipid-displacement-activation-mechanism induced by C16 in TRPV2: (i) C16 penetrates the interface between S4-S5 of one subunit and S5-S6 of a neighboring subunit from the intracellular side; (ii) C16 then destabilizes the interaction of the lipid within the vanilloid pocket, and causes the Y525 sidechain and backbone to flip towards the intracellular side; (iii) this removes the steric hindrance between S4-S5 linker and S5-S6 of adjacent subunits, shortens S6 at its intracellular end and displaces S5-S6 from the pore, thereby promoting the opening of the LG. Such a mechanism is also in agreement with what is known about ligand-induced structural shifts necessary for channel opening/inhibition of TRPV2 and other TRPV channels, such as TRPV1 and TRPV6[21].

Then what about the partially overlapping binding sites of C16 and PLG, with opposite effects on channel function (Fig. 7)? In agreement with this notion, TRPV2$_{apo-2}$ and TRPV2$_{PLG}$ are alike overall (Supplementary Table 2). However, we observe significant local differences at the end of helix S6 that assists in forming the LG (Fig. 7d–f), and a movement of S4-S5 towards the central pore (S5-S6 of the adjacent monomer), between TRPV2$_{PLG}$ and TRPV2$_{C16-2}$ (Fig. 7e, f). Thus, these relatively small but distinct differences likely modulate TRPV2 activity. This may inspire future ligand modification efforts, as the effect of compounds may be altered using subtle changes of their structures.

## Cryo-EM structure of TRPV2 in the presence of probenecid

Incubation of TRPV2 with probenecid resulted in a single structure, TRPV2$_{probenecid}$, with an overall resolution of 3.5 Å, with a somewhat flat shape of the cryo-EM density of the sidechains in the TM domain suggesting higher intrinsic flexibility (Fig. 8a). The overall channel structure is maintained, and pore width calculations suggest that the ion-conducting pathway is less obstructed than in TRPV2$_{apo-1}$ but not as wide as in TRPV2$_{open}$ (Fig. 8b). Likely due to the somewhat lower resolution of TRPV2$_{probenecid}$, we were unable to identify a non-protein density representing the ligand, however, structural rearrangements observed in TRPV2$_{probenecid}$ compared to TRPV2$_{C16}$, TRPV2$_{CBD}$ and TRPV2$_{apo}$ strongly indicate probenecid-driven changes. Most notable

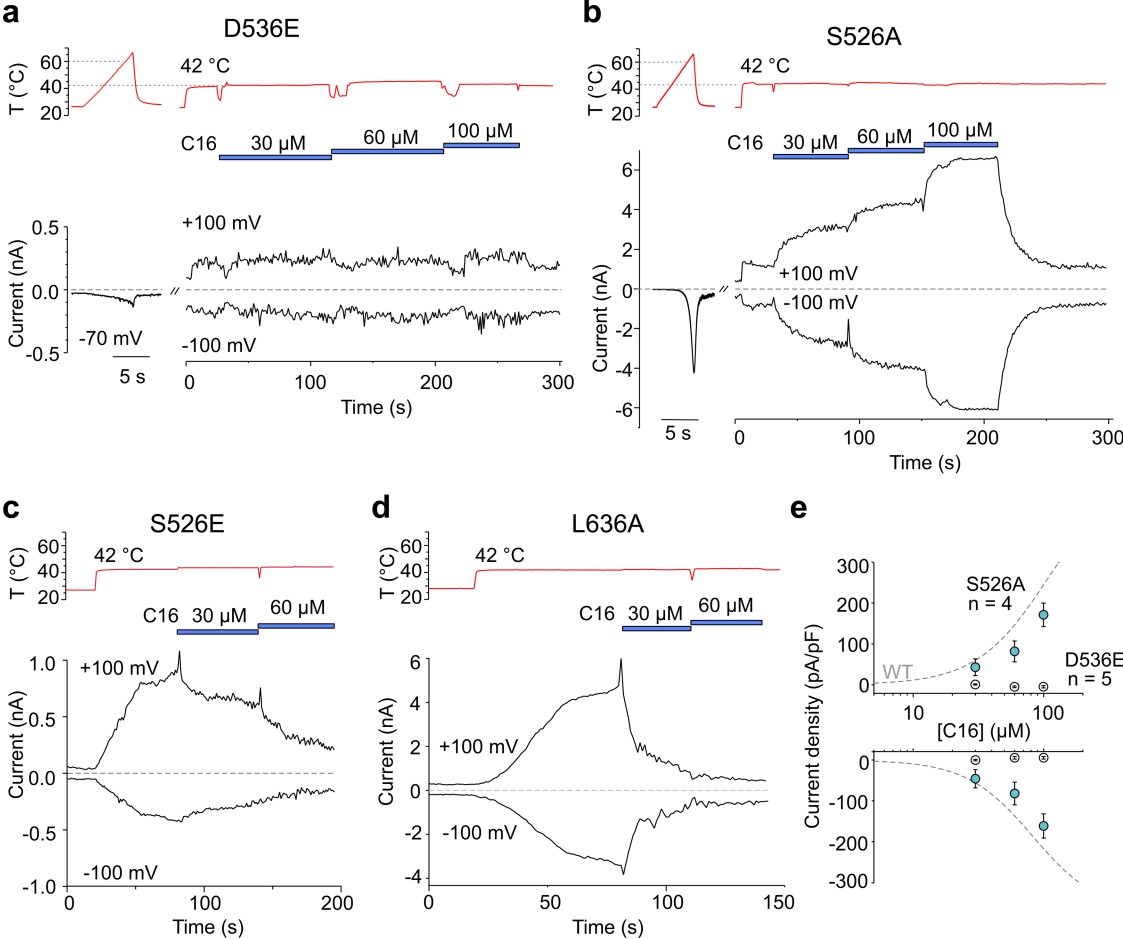

**Fig. 6 | Functional validation of the C16 binding site.** Representative whole-cell currents of D536, S526 and L636 rat TRPV2 mutants expressed in HEK293T cells elicited by heat applied in control extracellular solution and then by C16 applied sequentially at concentrations of 30, 60 and 100 μM (marked by horizontal blue bars) at 42 °C. **a** C16 did not evoke outward and inward currents in the D536E mutant. **b** The mutant S526A retains the sensitivity to C16. **c**, **d** As shown by representative traces for S526E ($n = 8$ independent experiments) and L636A ($n = 7$ independent experiments) mutants, C16 inhibited the currents caused by a rise in the temperature from 25 to 42 °C. **e** Concentration-response relationships for C16 activation of the D536E and S526A mutants. Data are shown as means ± SEM from five (D536E) and four (S526A) independent experiments. Source data are provided as a Source Data file.

is the finding that the vanilloid pocket is vacant as shown for TRPV2$_{open}$[46] (Fig. 8a). The electrostatic surface of TRPV2$_{probenecid}$ vanilloid pocket is similar to that of the two structures generated in the presence of C16 (Supplementary Fig. 19). However, the pocket is smaller due to the reorientation of the R517, Q663 and Q530 sidechains, too small to accommodate a lipid headgroup or the negative carboxylic acid group of CHS (Fig. 8c, d, Supplementary Fig. 19). This supports the notion that probenecid may negatively affect lipid binding. Thus, even though we do not clearly detect a bound probenecid molecule in our structure, our data point towards probenecid being able to release lipid from the vanilloid pocket.

To understand the effect of probenecid on TRPV2 in more detail, we aligned TRPV2$_{apo-1}$ with TRPV2$_{probenecid}$ (Fig. 8e–g). Interestingly, the S1–S4 bundle and the S5-S6 domain shift towards each other in the TRPV2$_{probenecid}$ structure, with larger movements on the intracellular side than the extracellular side (Fig. 8f, g). Similar but smaller movements are also observed when comparing TRPV2$_{apo-1}$ and TRPV2$_{C16-2}$ (Fig. 7a–c). Therefore, removal of the lipid from the vanilloid pocket situated between S1–S4 bundle and S5-S6 pore may allow a tighter arrangement of the two domains and thus facilitate pore opening. Collectively, we identified probenecid-induced opening of TRPV2, although not at the same magnitude as what was observed in the TRPV2$_{open}$ structure. This suggests that the TRPV2$_{probenecid}$ structure represents a pre-open or sensitized state.

## The combination of C16 and probenecid causes additional conformational changes

We obtained two cryo-EM structures of TRPV2 using a sample treated with both C16 and probenecid, TRPV2$_{C16+pro-1}$ and TRPV2$_{C16+pro-2}$, determined at 3.3 Å and 3.4 Å overall resolution (Fig. 9a). The local resolution of these cryo-EM structures differed substantially compared to those of TRPV2 for each ligand alone (Supplementary Fig. 20). Both structures were best resolved with lower (C1) symmetry, in contrast to the other C4-symmetric structures. Overall, TRPV2$_{C16+pro-1}$ was relatively poorly resolved, especially the ion-conducting pore (Supplementary Fig. 21), which may relate to flexibility caused by the binding of two ligands. The SF and LG in TRPV2$_{C16+pro-1}$ are wider than in TRPV2$_{C16+pro-2}$ (Fig. 9a). Moreover, the SF is narrower in both TRPV2$_{C16+pro-1}$ and TRPV2$_{C16+pro-2}$ than in TRPV2$_{C16-1}$, and broader than in TRPV2$_{probenecid}$ and TRPV2$_{C16-2}$ (Fig. 9a). This could indicate a synergistic effect of C16 and probenecid relative to probenecid alone, or spontaneous opening of SF. Another potential synergistic effect of probenecid and C16 is an overall wider LG in TRPV2$_{C16+pro-1}$ compared to the structures obtained in the presence of probenecid or C16 alone (Fig. 9a). We did not observe density for probenecid or lipids in either of the two TRPV2$_{C16+pro}$ structures, and the transmembrane domain configuration suggests that the vanilloid pocket lipid was released from TRPV2 treated with both probenecid and C16, as shown for probenecid alone. Interestingly, we identified a density, which could

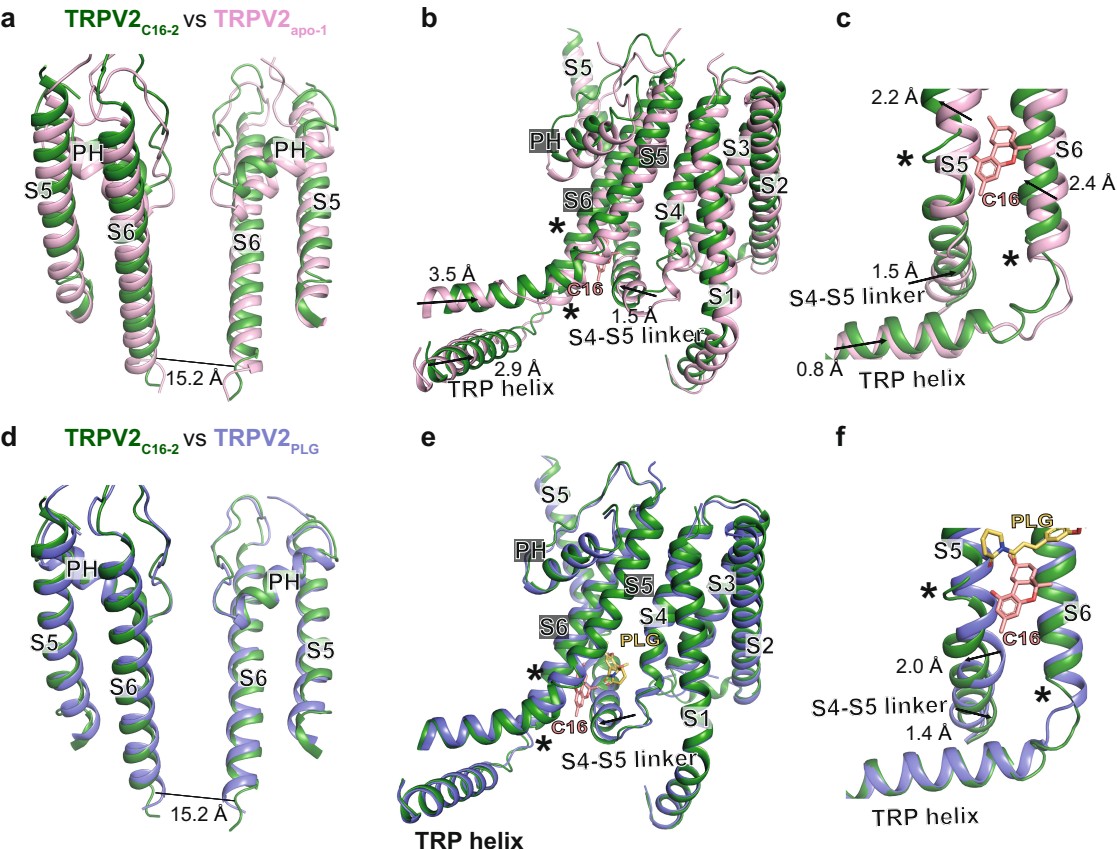

**Fig. 7 | C16-induced conformational changes.** Overlay of the (**a-c**) TRPV2$_{C16-2}$ (dark green) and TRPV2$_{apo-1}$ (PDB-ID: 6U84, pink) and (**d-f**) TRPV2$_{C16-2}$ (dark green) and TRPV2$_{PLG}$ (PDB-ID: 6WKN, purple) structures based on global tetramer alignment. **a-c** Membrane views of TRPV2$_{C16-2}$ and TRPV2$_{apo-1}$ showing (**a**) two opposite S5-PH-S6 units, (**b**) a single subunit (S1-S5) and an adjacent subunit (showing S5-PH-S6-TRP helix), and (**c**) the local changes around the suggested C16 binding site. **d-e** Membrane views similar to **a-c** comparing TRPV2$_{C16-2}$ and TRPV2$_{PLG}$. Arrows indicate movements of TRPV2$_{C16-2}$ relative to TRPV2$_{apo-1}$ and TRPV2$_{PLG}$, respectively. Local conformational changes are highlighted with stars. C16 is displayed as pink sticks in **b**, **c** and **e**, **f**.

possibly be C16, in the C16 binding site in-between two of the monomers (chains C and D) of TRPV2$_{C16+pro-2}$ (Fig. 9b). Therefore, to explore any synergistic effect of probenecid and C16 on the TRPV2 structures, we compared chains C and D in TRPV2$_{C16+pro-2}$ to TRPV2$_{probenecid}$ and TRPV2$_{C16-2}$. The structural differences between TRPV2$_{C16+pro-2}$ and TRPV2$_{probenecid}$ are mainly at the C16 binding site. The intracellular end of S5 is displaced away from the pore, and the TRP-helix shifts away from the pore (Fig. 9b). In addition, the loop connecting S6 to the TRP-helix is stretched. Collectively, these structural changes affect the LG of TRPV2$_{C16+pro-2}$ compared to TRPV2$_{probenecid}$ (Fig. 9b). Movement of the latter is still prominent even compared with TRPV2$_{C16-2}$, which would fit with a synergistic effect of C16 and probenecid (Fig. 9c). However, we may have captured TRPV2$_{C16+pro-2}$ in either a pre-open or desensitized state as the LG is not as open as in TRPV2$_{C16-2}$. Interestingly, the absence of a CHS-like molecule in TRPV2$_{C16+pro-2}$ seems to affect the hydrogen-bond triad network formed by Y544, T604 and Y629, previously suggested to play an essential part in the transformation between closed and open states of TRPV2[61], differently than in TRPV2$_{C16-2}$ (Fig. 9c). Thus, the altered positions of Y544, T604 and Y629 in TRPV2$_{C16+pro-2}$ may restrict the motion of S6 towards the extracellular side, thereby limiting the opening of the LG compared to TRPV2$_{C16-2}$. Taken together, we favor a model where probenecid can destabilize the protein in the LG region, as we observed in the presence of probenecid alone, and still permit binding of C16, as additional changes of the structure emerge from the C16 binding site, although we observe density possibly corresponding to C16 at only one of four binding sites. Furthermore, we observe that the density of one subunit displays smearing and relatively low quality compared to the other

three subunits in both TRPV2$_{C16+pro}$ structures, strongly suggesting high structural flexibility in the samples being treated with two ligands.

In summary, several substances modulate TRPV2 activity (Fig. 10) and among the cannabinoids, the phytocannabinoids $\Delta^9$-THC and CBD have good potency and efficacy[20]. However, structural evidence of a direct interaction between TRPV2 and cannabinoids has so far only been demonstrated for CBD, which binds to residues in a hydrophobic pocket between the S5 and S6 helices of adjacent subunits[27]. Notably, the chemical structure of CBD is different from $\Delta^9$-THC and other TRPV2-active $\Delta^9$-THC analogs. Here, we used C16, with a $\Delta^9$-THC scaffold that lacks CB1 and CB2 receptor activity[32], and propose another site of action for such cannabinoids on TRPV2 (Fig. 10). This site is within the vanilloid pocket and partially overlaps with that of the TRPV2 inhibitor PLG[28]. Intriguingly, as has been observed for other ligands and other TRPV channels, the molecular effects of C16 and PLG likely converge by modulating the binding of regulatory lipids.

As with the previously identified CBD site, the C16 binding site may be conserved within the TRPV family as we found that C16 can also activate TRPV1. Importantly, the effect of C16 on TRPV2 was substantially potentiated by probenecid, the binding site of which remains to be identified. However, we found structural evidence that the interaction of probenecid with TRPV2 leads to the displacement of phospholipid from the vanilloid pocket, and that probenecid alone induces conformational changes distinct from those observed in combination with C16.

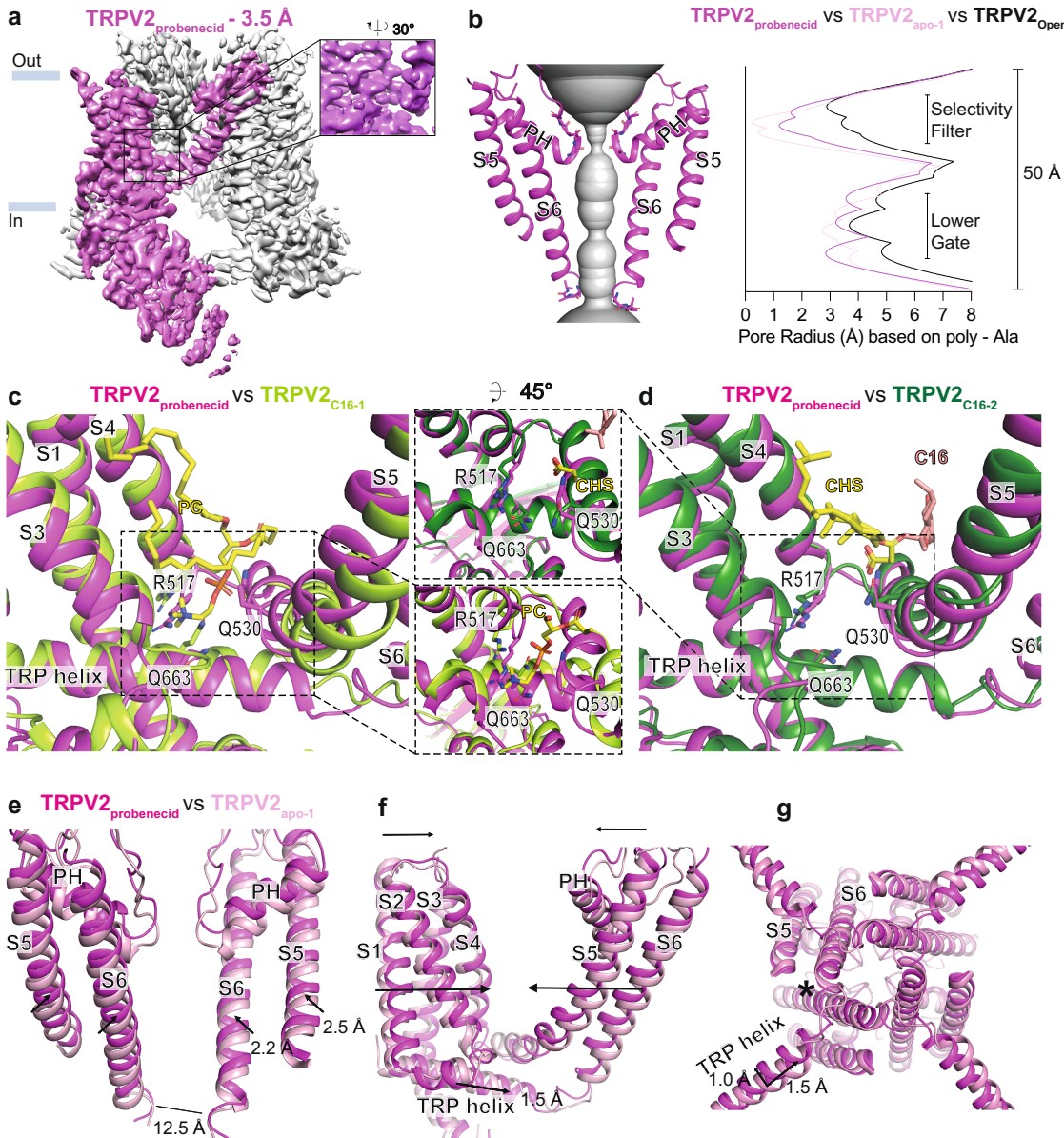

**Fig. 8 | Probenecid-induced conformational changes. a** Cryo-EM maps of TRPV2$_{probenecid}$ at 3.5 Å overall resolution with one subunit shown in magenta and the remaining subunits in grey. **b** The profile of the ion permeation pathway, displayed as grey space-filling model, (left) and the corresponding calculated pore radius as a function of the distance along the ion conduction pathway (right) for TRPV2$_{probenecid}$ (magenta). The pore radii of TRPV2$_{OPEN}$ (PDB-ID:6BO4; black line) and TRPV2$_{apo-1}$ (PDB-ID: 6U84; pink line) are also included as references. **c, d** Overlay of (**c**) TRPV2$_{probenecid}$ (magenta) and TRPV2 $_{C16-1}$ (light green) and (**d**) TRPV2$_{probenecid}$ TRPV2$_{C16-2}$ (dark green) structures based on the alignment of a single subunit. Residues of interest are labelled and represented as sticks. Also shown are PC (**c**) and CHS (**d**) as yellow sticks, C16 (**d**) as pink sticks. **e-g** Overlay of globally aligned TRPV2$_{probenecid}$ (magenta) and TRPV2$_{apo-1}$ (pink). **e** Membrane view comparisons of opposite pore-forming subunits. **f** Membrane view of the trans-membrane region of a single subunit illustrating the differences in the vanilloid pocket region. **g** View of the pore from the intracellular side of the membrane. PH indicates pore helix, and arrows indicate structural movements of TRPV2$_{probenecid}$ in relation to TRPV2$_{apo-1}$. The length of the arrow displays the varying degrees of structural changes. Star indicates local conformational changes.

Taken together, we have identified a new cannabinoid binding site in TRPV2 that is under the influence of allosteric control by probenecid. This adds further insight into the molecular mechanism by which ligands modulate TRPV2 to fully understand the relevance of this channel in normal and pathophysiology, and a framework for the development of refined substances of potential therapeutic value.

## Methods

### Test compounds
Δ⁹-Tetrahydrocannabiorcol (C16) was synthesized as previously described[32] and by ReadGlead AB (Lund, Sweden). Probenecid was

kindly supplied as a sodium salt by Dr Johan Raud, Karolinska Institutet and Dr Ulf Nilsson (Lund University). Allylisothiocyanate from mustard oil (MO), capsazepine, HC030031, ruthenium red and A23187 (Sigma-Aldrich). Capsaicin (Tocris Bristol, UK). Ionomycin (Invitrogen). Probenecid and ruthenium red were dissolved in saline. All other test compounds were dissolved in ethanol or DMSO. In all assays, the final concentration of ethanol and/or DMSO did not exceed 1%.

### Cell culture and transfections for Ca²⁺ imaging and planar patch-clamp electrophysiology
Human embryonic kidney cells (HEK293T, ECACC, 12022001) were grown in Dulbecco's modified Eagle's medium (DMEM, Life

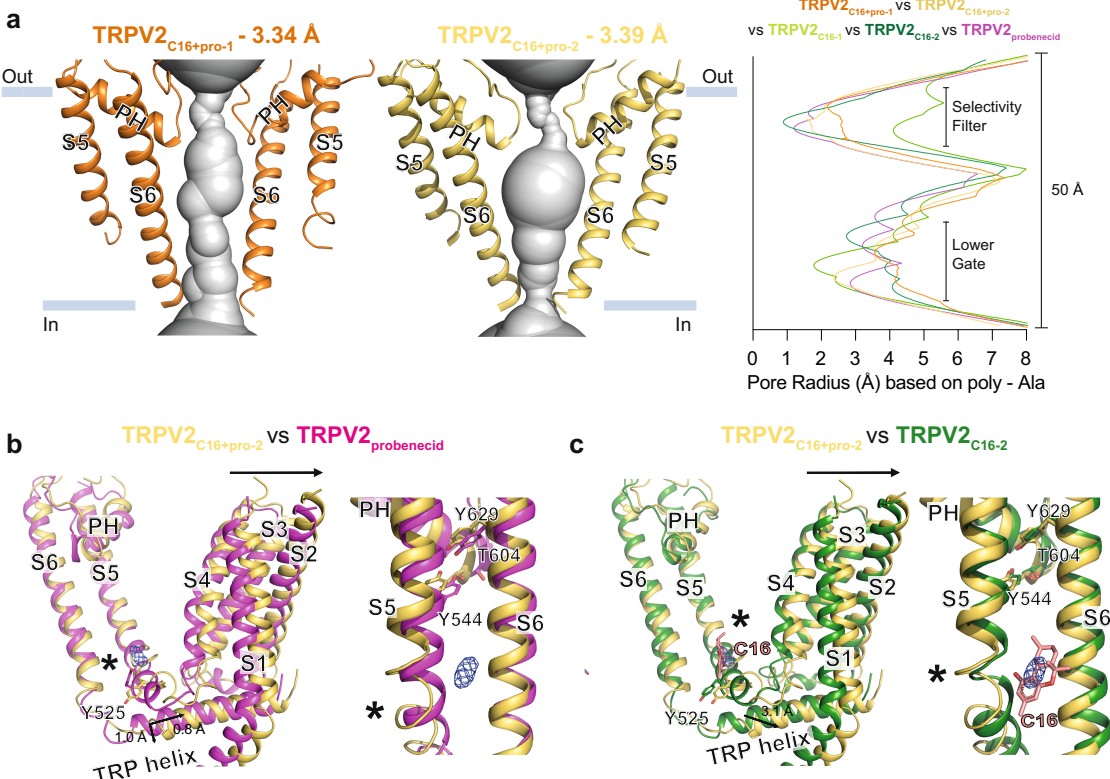

**Fig. 9 | The combined effect of C16 and probenecid on TRPV2. a** Profile of the ion permeation pathway, displayed as gray space-filling models, for TRPV2$_{C16+pro-1}$ (orange) and TRPV2$_{C16+pro-2}$ (yellow). The corresponding calculated pore radius as a function of the distance along the ion conduction pathway is also shown for TRPV2$_{C16-pro-1}$ (orange), TRPV2$_{C16-pro-2}$, (yellow), TRPV2$_{C16-1}$, (light green), TRPV2$_{C16-2}$ (dark green) and TRPV2$_{probenecid}$ (magenta). **b, c** Overlay of TRPV2$_{C16+pro-2}$ with either TRPV2$_{probenecid}$ or TRPV2$_{C16-2}$ structures based on global structural alignments. Membrane views of subunits of the vanilloid pocket (left) and S5-PH-S6 domain (right) are shown (PH designate pore helix). Arrows indicate structural movements of TRPV2$_{C16+pro-2}$ in relation to either TRPV2$_{probenecid}$ or TRPV2$_{C16-2}$. The blue mesh indicates cryo-EM density that may relate to C16. Residues of interest are shown as sticks. Local conformational changes are highlighted with stars.

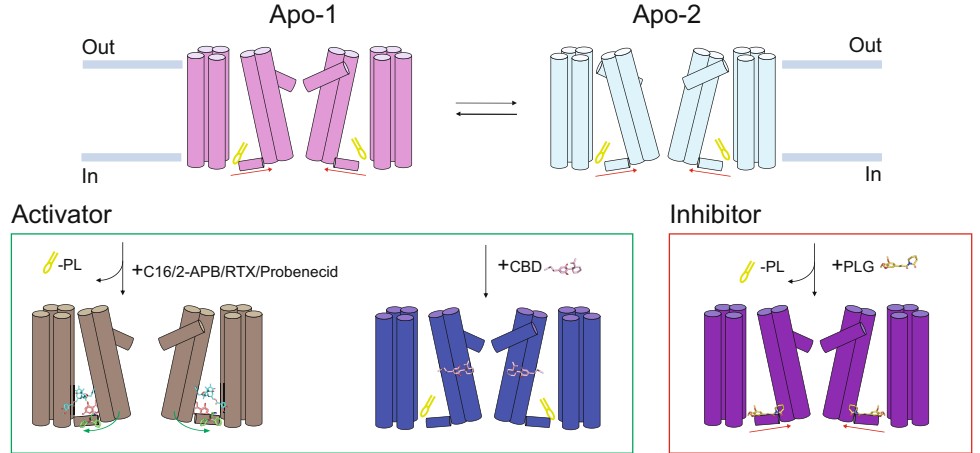

**Fig. 10 | Activation and inhibition of TRPV2 by various ligands.** The vanilloid pocket of ligand-free TRPV2 is occupied by lipids and demonstrates auto-stimulation of the selectivity filter (SF, or upper gate), as observed in-between two TRPV2 apo structures (TRPV2$_{apo-1}$, PDB-ID: 6U84 and TRPV2$_{apo-2}$, PDB-ID: 6U86). Two ligand-dependent activating mechanisms of TRPV2 are displayed. The presence of C16, probenecid, 2-APB (PDB-ID: 7T37) or RTX (PDB-ID: 6OO7) at different locations in TRPV2 displace the lipid from the vanilloid pocket (note: RTX only activates a modified TRPV2 form). In contrast, CBD (PDB-ID: 6U8A and 6U88) is unable to repel the lipid from the vanilloid pocket. PLG (PDB-ID: 6WKN) inhibition is achieved via the replacement of the lipid in the vanilloid pocket. The arrows indicate approximate molecular shifts.

Technologies) supplemented with 10% fetal calf serum (Sigma-Aldrich), 1% GlutaMAX (Life Technologies) and L-glutamine-penicillin-streptomycin (Sigma-Aldrich; 2 mM, 100 IU ml$^{-1}$, 100 µg ml$^{-1}$ respectively). Stable cell lines expressing rat, mouse and human TRPV2 were generated by transfecting TRPV2 expression constructs into HEK293T cells using FUGENE (Promega) as a carrier according to manufacturers' protocol followed by selection for G418 resistance. cDNA encoding rat and mouse TRPV2 was inserted in a pcDNA3 vector,

whereas the human TRPV2 was in an pCi-neo/IRES-GFP vector (kindly provided by Dr Michael Caterina at Johns Hopkins University, USA, and Dr Bernd Nilius at KU Leuven University, Belgium). After individual clones were selected by limited dilution, cells were grown in HEK293T culture medium supplemented with 0.4 mg ml⁻¹ G418 (Sigma-Aldrich). human TRPA1- and human TRPV1-HEK293Trex cells (kindly provided by AstraZeneca, Södertälje, Sweden) were grown in DMEM supplemented with 10% fetal calf serum, 1% GlutaMAX, 10 μg ml⁻¹ blasticidin S hydrochloride (Invitrogen) and 500 μg ml⁻¹ hygromycin B (Invitrogen). human TRPA1- and human TRPV1-expressing cells were induced with 0.1 μg ml⁻¹ tetracycline (BD Bioscience) overnight and used for experiments the following day. Untransfected HEK293Trex cells (control) were grown in a similar medium containing only 5 μg ml⁻¹ blasticidin S hydrochloride and no hygromycin B. Rat basophilic leukemia cells (RBL-2H3, ATCC, CRL-2256) were grown in Minimum Essential Medium α with GlutaMAX (Life Technologies) supplemented with 15% fetal calf serum and glutamine-penicillin-streptomycin. The human mast cell line (HMC-1.2; kindly provided by Dr Joseph H. Butterfield, Mayo Clinic Hospital, Rochester, MN, USA) was cultured in Iscove's modified Dulbecco's medium (IMDM) supplemented with 10% fetal calf serum (FCS), 2 mM L-glutamine, 100 IU ml⁻¹ penicillin, 50 mg ml⁻¹ streptomycin, 1.2 mM α-thioglycerol.

### Fluorometric Calcium imaging

Measurement of $[Ca^{2+}]_i$ was performed as previously described[35] and as follows. HEK293T, RBL-2H3 and HMC-1.2 cells were seeded in poly-D-lysine-coated 96-well black-walled plates (Costar) or 8-well slides (Ibidi) and on the following day loaded for 1 h at 37 °C with 1-2 μM Fura-2 AM (Invitrogen), 2 mM probenecid (Sigma-Aldrich) and 0.02% pluronic acid (Invitrogen) in a physiological buffer containing (in mM) NaCl 140, KCl 5, $CaCl_2$ 2, $MgCl_2$ 1, HEPES 10, Glucose 10 (pH 7 4). After exchanging the loading buffer, changes in $[Ca^{2+}]_i$ were determined in a Flexstation 3 (Molecular Devices) or with an Olympus IX70 microscope connected to a digital camera (Hamamatsu) controlled by the software program SimplePCI 6 (Hamamatsu). Basal emission (510 nm) ratios with excitation wavelengths of 340 and 380 nm were measured and changes in dye emission (Δratios) were determined at various times after drug application.

Data analysis was done in SoftMax Pro Analysis (Molecular Devices) or data were exported with extrapolation from SoftMax Pro Analysis into Excel (Microsoft, version 15, 2013). Δratios were given as peak values or Area Under the Curve (AUC) values, and corrected for the average Δratio for probenecid + vehicle when probenecid was applied prior to the second agonist application. In some experiments, Δratios for agonists were normalized to the average Δratio for ionomycin following vehicle or probenecid + vehicle application, whereas for the RBL-2H3 cell experiments the Δratios were normalized to the average peak response for 3 mM probenecid in presence of vehicle. The concentration-response relationship was analyzed by fitting the Δratios to a four-parameter logistic equation in GraphPad Prism 9.1 (GraphPad Software).

### Whole-cell patch-clamp electrophysiology

**Cell culture, constructs and transfection.** Human embryonic kidney 293 T (HEK293T; ATCC, CRL-1573) cells were cultured in Opti-MEM I media (Invitrogen, Carlsbad, CA, USA) supplemented with 5% fetal bovine serum. The magnet-assisted transfection (IBA GmbH, Gottingen, Germany) technique was used to transiently co-transfect the cells in a 15.6 mm well on a 24-well plate coated with poly-L-lysine and collagen (Sigma-Aldrich, Prague, Czech Republic) with 200 ng of GFP plasmid (TaKaRa, Shiga, Japan) and with 300 ng of cDNA plasmid encoding wild-type or mutant rat TRPV2 (in the pcDNA3.1 vector, a kind gift from Dr. Francois Rassendren, Université de Montpellier, France). The cells were used 24–48 h after transfection. At least three independent transfections were used for each experimental group. The wild-type channel was regularly tested in the same batch as the

mutants. The mutants were generated by PCR using specific primers ordered from Sigma-Aldrich (Prague, Czech Republic) and a Quik-Change II XL Site-Directed Mutagenesis Kit (Agilent Technologies, Santa Clara, CA, USA), and confirmed by DNA sequencing (Eurofins Genomics, Ebersberg, Germany).

**Whole-cell electrophysiology and temperature stimulation.** Whole-cell membrane currents were filtered at 2 kHz using the low-pass Bessel filter of the Axopatch 200B amplifier and digitized (10 kHz) using a Digidata 1440 unit and pCLAMP 10 software (Molecular Devices). Patch electrodes were pulled from borosilicate glass and heat-polished to a final resistance between 3 and 5 MΩ. Series resistance was compensated by at least 60%. Extracellular solution was $Ca^{2+}$-free and contained: 150 mM NaCl, 5 mM EGTA (ethylene glycol-bis (β-aminoethyl ether)-N,N,N′,N′-tetraacetic acid), 10 mM 4-(2-Hydroxyethyl)piperazine-1-ethanesulfonic acid (HEPES), 10 mM glucose, pH 7.4 was adjusted by NaOH. Intracellular solution contained 140 mM CsCl, 4 mM $MgCl_2$, 10 mM HEPES and 10 mM EGTA, adjusted with CsOH to pH 7.4. The I−V relationships were recorded by using voltage ramps from −100 to +100 mV (0.4 V/s delivered every 1 s from a holding potential of 0 mV) or at a steady-state holding potential of −70 mV. Only one recording was performed on any one coverslip of cells to ensure that recordings were made from cells not previously exposed to heat. A system for fast heating of solutions superfusing isolated cells under patch-clamp conditions was used as described previously[62].

### Planar whole-cell patch-clamp electrophysiology

HEK293T and RBL-2H3 cells were harvested using trypsin/EDTA (Life Technologies), washed and resuspended in extracellular solution containing (in mM): NaCl 140, KCl 5, $MgCl_2$ 1, $CaCl_2$, 2, D-glucose 10, HEPES 10 (pH 7.4). Whole-cell recordings were performed with an automated patch-clamp platform using a chip size of 2-3.5 MΩ (Port-a-Patch, Nanion Technologies GmbH) and an EPC10 amplifier (HEKA). The intracellular solution contained (in mM): CsCl 50, CsF 60, NaCl 10, EGTA 20, HEPES 10, pH 7.2 (CsOH), ~285 mOsmol. Series resistance was <15 MΩ and compensated ~70%. The holding potential was −60 mV, and the liquid junction potential was not compensated. Whole-cell currents were low-pass filtered at 2.9 kHz, digitized at 10 kHz and recorded on a computer hard disk using the EPC10 amplifier and PatchMaster software (v2x65, HEKA). Drugs were applied directly to the chip in small volumes. Peak membrane currents were analyzed using Clampfit 10.3 (Molecular Devices).

### Planar lipid bilayer patch-clamp electrophysiology

These experiments were performed as originally described[35] and as follows. Detergent purified rat TRPV2 were reconstituted either into preformed planar bilayers or giant unilamellar vesicles (GUVs), composed of 1,2-diphytanoyl-sn-glycero-3-phosphocholine (Avanti Polar Lipids) and cholesterol (Sigma-Aldrich) in a 9:1 ratio and produced by using the Vesicle Prep Pro Station (Nanion Technologies, Germany). Planar lipid bilayers were formed by pipetting 5 μl of either empty GUVs or protein reconstituted into GUVs on patch-clamp chips (1-2 μm, 3.5–5 MΩ resistance) which were mounted on a recording chamber. Following giga Ohm seal formation, single-channel activity was recorded using the Port-a-Patch (Nanion Technologies) at both positive and negative test potentials in a symmetrical $K^+$ solution adjusted to pH 7.2 with KOH and containing (in mM): 50 KCl, 10 NaCl, 60 KF, 20 EGTA, and 10 HEPES. Signals were acquired with an EPC 10 amplifier (HEKA) and the data acquisition software Patchmaster (v2x65, HEKA) at a sampling rate of 50 kHz. The recorded data were digitally filtered at 3 kHz. Electrophysiological data were analyzed using Clampfit 9 (Molecular Devices) and Igor Pro (Wave Metrics software). Data were filtered at 1,000 and 500 Hz low-pass Gaussian filter for analysis and traces, respectively. The single-channel mean conductance (Gs) was obtained from a Gaussian fit of all-points amplitude histograms. The

single-channel mean open probability (Po) was calculated from time constant values, which were obtained from exponential standard fits of dwell-time histograms.

## Mast cell physiology

RBL-2H3 cells were cultured in Dulbecco's Modified Eagle Medium (DMEM, Sigma) containing 10% filtered fetal calf serum (FCS, Gibco), 100 IU ml$^{-1}$ penicillin G and 100 µg ml$^{-1}$ streptomycin (Sigma). At the delivery unit, umbilical cord blood was taken from healthy mothers with no complications in connection to the delivery and with informed consent obtained from all participants. The cord blood collection, since the donors are unidentifiable for the investigators, has been granted exemption from requiring ethics approval (Regional Ethics Review Board in Stockholm). Cord blood-derived mast cells (CBMCs) were derived and then maintained as previously described[63,64]. RBL-2H3 cells were seeded at a concentration of $0.5 \times 10^6$ cells ml$^{-1}$ and incubated overnight whereas CBMCs were seeded at a concentration of $0.5 \times 10^6$ cells ml$^{-1}$ prior activation. For activation, cells were washed with phosphate-buffered saline (PBS) and then incubated with compounds at various concentrations (probenecid at 1 mM or 10 mM, C16 at 100 µM, ruthenium red at 10 µM, and calcium ionophore A23187 at 1 µM) dissolved in piperazine-N,N'-bis(2-ethanesulfonic acid) (PIPES, Sigma) buffer (pH 7.4) supplemented with 0.2% bovine serum albumin (BSA, Sigma). Control contained only vehicle (PIPES + DMSO 0.1%). After 30 min of activation, supernatants were spun down and collected. Histamine release was measured using a histamine release test where histamine binds tightly to a glass fiber matrix which is then released and analyzed with the histareader at RefLab (RefLab, Copenhagen, Denmark). Histamine release was normalized against the release of histamine evoked by A23187, where 245 (median), 143–914 (range) ng ml$^{-1}$ of histamine was released for RBL-2H3 cells and 260 (median), 104–491 (range) ng ml$^{-1}$ of histamine was released for CBMCs.

## Expression of rat TRPV2

Rat TRPV2, C-terminally fused to GFP and a His$_8$-tag, was expressed in *Pichia pastoris* strain SMD1168 (Invitrogen, C17500). Cells were grown in 500 ml YNB glycerol medium supplemented with biotin at 30 °C for 24 h. Cells were then harvested by centrifugation (3100 × g for 10 min, 18 °C) and transferred into YNB cultures (cells from 250 ml preculture per liter main culture) with 0.5% methanol for protein expression induction. Cells were grown at 30 °C for ~10 h, then another dose of methanol (0.5%) was added. Cells were then grown for ~14 h. Finally, cells were harvested by centrifugation (3100 × g for 10 min, 4 °C) and washed once with deionized water. Pellets were flash frozen in LN$_2$ and stored at −80 °C until further use.

## Purification of rat TRPV2 for planar lipid bilayer electrophysiology

Frozen cells were resuspended in pre-cooled lysis buffer (50 mM Tris-Cl pH8, 1 mM EDTA, 1 mM DTT, 1 mM benzamidine, 1 mM PMSF, protease inhibitor tablet (Roche complete), supplemented with RNaseA and DNaseI) and lysed with a French press (seven passes, 30 bar). Lysate was spun down at 4800 × g for 10 min, 4 °C. The supernatant was then ultracentrifuged (at 235.000 × g for 60 min, 4 °C). Supernatant was discarded and pellets resuspended in 20 mM Tris, pH 7.5, 150 mM NaCl, supplemented with protease inhibitors and reducing agents. Total membrane protein concentration was determined with the BioRad RC Assay. Membranes were flash frozen in LN$_2$ and stored at −80 °C until further use.

For protein purification, membranes were thawed on ice and resuspended in solubilization buffer (20 mM NaPi pH 7.4, 500 mM NaCl, 10% v/v glycerol, 20 mM imidazole pH 8 supplemented with 1 mM PMSF, 1 mM benzamidine and 1 mM DTT) to yield a final concentration of total membrane protein of 15 mg ml$^{-1}$. Protein was solubilized with 1% DDM (n-Dodecyl-β-D-Maltoside) for 1.5 h, at 4 °C using mild stirring. The

suspension was then centrifuged at 235.000 × g for 30 min, 4 °C. The supernatant was loaded on a HisTrap NiSepharose column (Cytiva) for purification via the C-terminal His$_8$-tag. Unbound sample was washed away with two column volumes washing buffer (20 mM NaPi pH 7.4, 500 mM NaCl, 20 mM Imidazole pH 8, 0.03% DDM). Next, the column was washed with a four-column volumes gradient from 0 to 150 mM imidazole and then eluted with a four-column volume gradient from 150 to 500 mM imidazole. The purity of the eluted protein was analyzed using SDS-PAGE. Protein was then concentrated for size exclusion chromatography carried out on a Superose 6 column (GE Healthcare) in 20 mM Tris pH 8, 150 mM NaCl, 1 mM DTT, 0.03% DDM. Purified protein was aliquoted and flash-frozen in LN$_2$.

## Purification of rat TRPV2 for cryo-EM

Before purification, 20 g cells were thawed on ice and resuspended in 20 ml lysis buffer (25 mM imidazole pH 7.5, 1 M NaCl, 10% glycerol, 1 mM EGTA, 1 mM EDTA) supplemented with 1 mM DTT and protease inhibitors (1 µg ml$^{-1}$ leupeptin, pepstatin, and chymostatin, and 1 mM PMSF). Then the resuspended cells were split into two 50-ml conical tubes with 10 ml in each. An equal volume of dry glass beads was added to each tube. Cells were disrupted and vortexed 1 min 8 times. The supernatant was collected, and the glass beads were washed 3 times with no more than 100 ml lysis buffer. The totally lysate was centrifuged at 4200× g for 20 min to remove unbroken cells. Then the supernatant was ultra-centrifuged at 142.000× g for 90 min. The pellet was resuspended with low salt buffer containing 50 mM Tris pH 8.0, 300 mM NaCl, 10% glycerol, 1 mM DTT and protease inhibitors as before. Subsequently, the membranes were solubilized in 1% DDM and 0.33% CHS by stirring for 2 h, at 4 °C. Solubilized membranes were collected by ultracentrifugation at 104.000× g for 30 min. Buffer-washed Ni-beads (3 ml) were added to the supernatant and rotated for 2 h. Additional imidazole (30 mM) was supplemented to prevent unspecific binding. The sample was loaded onto a gravity column and washed with 10 column volumes of high salt buffer (50 mM Tris pH 8.0, 30 mM imidazole, 800 mM NaCl, 10% glycerol, 0.4 mg ml$^{-1}$ DDM, and 0.04 mg ml$^{-1}$ CHS) and 10 column volumes of low salt buffer supplemented with 30 mM imidazole and 0.4 mg ml$^{-1}$ DDM, and 0.04 mg ml$^{-1}$ CHS. This was followed by 10 column volumes of low salt buffer supplemented with 100 mM imidazole and 0.4 mg ml$^{-1}$ glyco-diosgenin (GDN). TRPV2 protein was eluted from beads with 10 ml of low salt buffer supplemented with 200 mM imidazole and 0.4 mg ml$^{-1}$ GDN. The eluted protein was pooled and concentrated to almost 2-3 mg ml$^{-1}$ using 10 K kDa cutoff concentrator device. The concentrated protein was applied to a Superose 6 column equilibrated with SEC buffer containing 20 mM Tris 8.0, 150 mM NaCl, 0.4 mg ml$^{-1}$ GDN. Peak fractions were collected and concentrated to approximately 3 mg ml$^{-1}$. The purity of the eluted protein was analyzed using SDS-PAGE (Supplementary Fig. 22).

## Cryo-EM sample preparation and data collection

Prior to preparing cryo-EM grids, purified TRPV2 was incubated with the ligands C16 (50 µM) and probenecid (2 mM), either alone or in combination at room temperature for 15 min. Meanwhile, 200 mesh Quantifoil 2/2 grids were glowed with 5 mA for 30 s. 3.5 µl sample was loaded onto glow discharged grids at 4 °C and 100% humidity and plunge frozen in liquid ethane by Vitrobot (Thermo Fisher Vitrobot). Blotting and waiting time were set to 3 and 10 s, respectively. Cryo data were collected on 300 kV Krios microscope equipped with a Gatan K3 direct detector camera in counting mode (Supplementary Table 2). Forty-frame movies were collected with a total dose ranging from 48.4 to 57 e$^-$ Å$^{-2}$, and a pixel size of 0.82 to 0.8464 Å. Defocus range was set from −0.7 to −2.7 mm at 0.2 mm steps.

## Data processing

All cryo-EM data were processed using Cryosparc[65], and as outlined in Supplementary Figs. 23–25 and Supplementary Table 2. Firstly, the

data were pre-processed with motion correction via full frame and patch motion correction. Next, particle blob picking was performed, and particles were re-extracted with box size 360 pixels and applying local motion correction with dose weighting. Then, several rounds of 2D classification without reference was executed to remove defective particles from the auto-picked particle set. Finally, the classical cryosparc workflow, including ab-initial model reconstruction, iterative heterogeneous refinement and non-uniform refinement (applying C4 for symmetry for the two structures obtained in the presence of C16 and TRPV2$_{probenecid}$, and C1 symmetry for TRPV2$_{C16+pro}$) was followed to generate the final maps (Supplementary Table 2 and Supplementary Figs. 23–25).

## Model building and refinement
Cryo-EM structures of TRPV2$_{apo}$ (PDB-ID: 6U84 or 6U86)[27] were employed as an initial model, which was fitted into the cryo-EM maps using rigid body fitting in Chimera[66]. Multiple rounds of manual and automated refinement in Coot[67] and phenix_real_space_refine[68] were applied to generate the final model. This strategy was applied to generate all structures. The quality of final models was evaluated using MolProbity[69] and the final structure refinement statistics for all models are summarized in Supplementary Table 2. The local density for each structure is displayed in Supplementary Figs. 23–25.

## Statistics
SigmaPlot 10 (Systat Software Inc., San Jose, USA), GraphPad Prism 9.1. (GraphPad Software, La Jolla, CA), Igor Pro (6.0.6.0., Wave Metrics software) and CorelDraw X7 (Corel corporation, Ottawa, Canada) were used for statistical analysis and drawing of graphs. The level of statistical significance was set at $P < 0.05$. Student's paired or unpaired t-test, Welch's unpaired t-test, Wilcoxon matched-pairs test and repeated measures one-way ANOVA followed by Sidak's multiple comparisons test or Dunnett's post hoc comparison tests were used for analysis of statistical significance. Data are presented as the mean ± SEM; n indicates the number of separate/independent experiments examined. The electrophysiological data were analyzed using pCLAMP 9 and 10 (Molecular Devices).

## Reporting summary
Further information on research design is available in the Nature Portfolio Reporting Summary linked to this article.

## Data availability
The data that support this study are available from the corresponding authors upon reasonable request. Cryo-EM maps have been deposited in the Electron Microscopy Data Bank (EMDB) under accession codes EMD-14745 (TRPV2$_{C16-1}$), EMD-14746 (TRPV2$_{C16-2}$), EMD-14747 (TRPV2$_{probenecid}$), EMD-14749 (TRPV2$_{C16+pro-1}$), EMD-4748 (TRPV2$_{C16+pro-2}$). The atomic coordinates have been deposited in the Protein Data Bank (PDB) under accession codes 7JD (TRPV2$_{C16-1}$), 7ZJE (TRPV2$_{C16-2}$), 7ZJG (TRPV2$_{probenecid}$), 7ZJI (TRPV2$_{C16+pro-1}$), 7ZJH (TRPV2$_{C16+pro-2}$). The atomic coordinates and cryo-EM density maps of previously solved TRPV2 structures used in this study are available under accession codes 6U84 (TRPV2$_{apo-1}$) and EMD-20677 (TRPV2$_{apo-1}$), 6U86 (TRPV2$_{apo-2}$) and EMD-20678 (TRPV2$_{apo-2}$), 6BO4 (TRPV2$_{open}$) and EMD-7118 (TRPV2$_{open}$), 7XEM (TRPV2$_{CHL}$) and EMD-33156 (TRPV2$_{CHL}$), 7S89 (TRPV6) and EMD-24891 (TRPV6), 6U8A (TRPV2$_{CBD-1}$) and EMD-20686 (TRPV2$_{CBD-1}$), 6U88 (TRPV2$_{CBD-2}$) and EMD-20682 (TRPV2$_{CBD-2}$), 6WKN (TRPV2$_{PLG}$) and EMD-21705 (TRPV2$_{PLG}$), 6OO7 (TRPV2$_{RTX-C2}$) and EMD-20148 (TRPV2$_{RTX-C2}$), 7T37 (TRPV2$_{APB+CBD-active}$) and EMD-25650 (TRPV2$_{APB+CBD-active}$), 6LGP (TRPV3) and EMD-0882 (TRPV3), 6BWJ (TRPV2$_{RTX-Crystal}$) in the PDB and EMDB, respectively. Plasmids of the TRPV2 wild-type and mutants thereof investigated in this study are available from the corresponding authors upon request. The source data underlying Figures 1,2,3,6; and Supplementary Figures 1–6, 15,16 are available as a Source Data file. Source data are provided with this paper.

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

## Acknowledgements

This study was supported by the Swedish Research Council (2014–3801; P.M.Z. and 2020-01693; G.N. as well as 2012–2243 and 2016-04474; P.G.) and the Medical Faculty of Lund University – ALF (Dnr. ALFSKANE-451751; P.M.Z), Hjärnfonden/the Swedish Brain Foundation (FO2020-0188 and FO2022-0292; P.M.Z.), Stiftelsen Olle Engkvist Byggmästare (189–290; P.M.Z.), Albert Påhlssons stiftelse (P.M.Z.), Alfred Österlunds stiftelse (P.M.Z.); Knut och Alice Wallenbergs Stiftelse (2015.0131 and 2020.0194; P.G.), The Lundbeck Foundation (R133-A12689, R313-2019-774 and R346-2020-2019; P.G.), the Danish Council for Independent Research (9039-00273; P.G.) and the Crafoord Foundation (20170818; P.G.); the Czech Science Foundation (grant number 22-13750S; V.V.). Support from the DFG Cluster of Excellence EXC2051 Balance of the Microverse—Project-ID 390713860, the DFG Collaborative Research Center 1507 "Membrane-associated protein assemblies, machineries and supercomplexes" (Project-ID 450648163, P11 to U.A.H.) and the National Institutes of Health R01GM081340 (R.G.) is kindly acknowledged. We would like to acknowledge Benedikt Goretzki, Sabine Häfner and Michal Mitro for technical support. We would also like to thank Julian Conrad, Dustin Morado and Marta Carroni at SciLifeLab Stockholm as well as Tillmann Hanns Pape at the Core Facility for Integrated Microscopy (CFIM) at University of Copenhagen for assistance with the grid screening and Cryo-EM data collection. The Cryo-EM Swedish National Facility at SciLifeLab is funded by the Knut and Alice Wallenberg, Family Erling Persson and Kempe Foundations, SciLifeLab, Stockholm University and Umeå University. The Danish Cryo-EM Facility at CFIM, University of Copenhagen is supported by Novo-Nordisk Foundation grant id NNF14CC0001.

## Author contributions

C.S. and P.M.Z. conceptualized research; L.Zh., C.S., V.V., P.G., P.M.Z. directed research; L.Zh. (cryo-EM), C.S. (Fura-2, patch-clamp), K.W. (cryo-EM), L.M. (bilayer patch-clamp), M.E. (mast cell histamine release), G.N. (mast cell histamine release), P.G. (cryo-EM), P.M.Z. (Fura-2, patch-clamp, cryo-EM) designed research; L.Zh. (cryo-EM), C.S. (Fura-2, patch-clamp), L.Zi (patch-clamp), L.M. (bilayer patch-clamp), M.E. (mast cell histamine release) performed research; L.Zh., C.S., L.Zi, K.W., L.M., R.G., M.E., G.N., U.A.H., V.V., P.G., P.M.Z. analyzed data; R.G., U.A.H. contributed TRPV2 construct and purified protein; L.Zh., C.S., R.G., U.A.H., V.V., P.G., P.M.Z. wrote the manuscript. All authors read and commented on the manuscript.

## Funding

## Competing interests

The authors declare no competing interests.
