## [Peer Review File · Nature Communications]

Cannabinoid non-cannabidiol site modulation of TRPV2 structure and functionReviewers' Comments:

Reviewer #1:

Remarks to the Author:

Here the authors study the opening of TRPV2 by a cannabinoid compound, C16, and the TRPV2 agonist probenecid. The first part of the study focuses on the functional characterization of TRPV2 activation by the two ligands. The authors find that the effects of C16 and probenecid are synergistic, i.e. that application of one followed by application of the other produces a larger current than application of either one on its own. The authors surmise that the ligands have different binding sites and mechanisms of action.

The second part of the study aims to identify the ligand binding sites and their mechanisms of action via analysis of cryo-EM structures of rat TRPV2 in the presence of C16 and probenecid.

It is my opinion that there would be some interest in the data amongst the TRP channel community.

However, there are several issues that need to be addressed.

Major concerns:

1. I would like to see some more evidence that the C16/probenecid effect isn't due to use dependence in TRPV2. For example, how does TRPV2 respond to repeated exposures to C16 and probenecid, respectively? Does the EC50 for each change upon repeated exposure? And how do those values compare to the change in TRPV2 sensitivity to probenecid upon C16 exposure? Does C16 need to be present to see the increased effect of probenecid, and vice versa? Or, can the effect be seen after mere exposure to C16 / probenecid (i.e., one ligand is washed out before application of the second one)?

Please see the section on ligand cross-sensitization in Liu, Beiying, and Feng Qin. "Use Dependence of Heat Sensitivity of Vanilloid Receptor TRPV2." *Biophysical journal* vol. 110,7 (2016): 1523-1537. doi:10.1016/j.bpj.2016.03.005.

Please also see the study of the related TRPV3 channel: Liu B, Yao J, Zhu MX, Qin F. Hysteresis of gating underlines sensitization of TRPV3 channels. *J Gen Physiol*. 2011;138(5):509-520. doi:10.1085/jgp.201110689

2. Ligand binding sites need to be validated by electrophysiology. This is especially important because ligands are small, the densities shown in the figures are weak, and cryo-EM does not offer a way to unambiguously identify bound molecules. Also, the authors should show half maps for the ligand densities.

3. The same applies to the proposed mechanisms – these should be tested by mutagenesis and electrophysiology.

4. The authors seem to assign a site to CHS without any supporting evidence. The site is used in downstream mechanistic explanations. I would strongly urge the authors to tone this down, given that no evidence is provided for CHS binding in that position.

5. The structural alignments are confusing – it is difficult to discern movements of individual domains when no fixed point exists in the alignment. I would suggest remaking the figure 7 a-c with a better alignment of the two structures and re-evaluating the conformational changes.

6. The structural analysis lacks some quantification – RMSD values, angles, distances, etc.

7. C16 alone barely activates TRPV2 in HEK cells but in bilayers it causes a 0.5 open probability? How is this explained? How does probenecid + C16 affect P_o in bilayers? Also, even though data for the C16 effect on empty bilayers has been published before, this should be shown here too. It's an important control that needs to be presented.

8. The choice to analyze the pore opening by removing side chains is curious. The Calpha distances aren't necessarily a good measure of the extent of pore opening, so the argument that something is more or less open based on HOLE profiles calculated from polyaniline models is perhaps a little misleading. I would argue that the language here should not include terms "open" or "closed" but instead refer to Calpha distances, with a clear acknowledgement that side chains will contribute to the functional state of the pore. Also, given the high resolution of the structures obtained here, the figures should also include a HOLE profile of the structures with side chains so that there is no doubt about

what the actual data tells us.

9. The reconstructions all reach good overall resolutions. However, the local resolution plots are missing and should be included.

Reviewer #2:

Remarks to the Author:

The manuscript untitled: 'Cannabinoid non-cannabidiol site modulation of TRPV2 structure and function' submitted by Zhang et al, identified tetrahydrocannabinol as an agonist of TRPV2. Via cryo-EM studies the authors identified the interaction site of C16 on TRPV2, a site that was similar for the previously described ligand pocket for CBD and 2-APB. In addition, the authors describe a synergistic effect of C16 and probenecid in mouse and human TRPV2 and showed that C16 and probenecid have different structural effects on the channel pore.

The presented work is a nice follow up study of recently published SAR studies work on TRPV2 of Pumroy et al 2022 in which a ligand interaction site for TRPV2 was identified. However, the current study has a lack on mutagenesis as a proof-of-concept. This is important to validate the conclusions, in particular the part in which the interaction of TRPV2 with probenecid is described.

Major comments:

- Despite the many benefits of the cryo-EM technique and the availability of the protein structure, the proposed method also has its limitations, as the purified proteins are not located in the PM of the cell and are crystalized without any physiological membrane potential. This is an important factor as all TRP channels (including TRPV2) are all voltage dependent. The conclusions of the manuscript are based on comparison studies of new and previously published cryo-EM models. However, the novel findings should be supported by mutagenesis studies as a proof-of-concept. For example: (1) No strong evidence is provided that the vanilloid pocket region is occupied with lipids like phosphocholine or PI. This is based on earlier speculations, but no hard evidence is proven. The authors should provide additional evidence via mutagenesis studies and neutralization of the positive charges at position R517 and K531. (2) the identification of the C16 binding site. The authors identified a potential region where the C16 could interact with the channel. However, to further provide stronger evidence mutagenesis studies and functional read-outs are required in which a series of key residues (H521 or Y525) should be neutralized as a proof of concept.
- In a recent study of Pumroy et al, 2021 Nature Communications evidence was provided for a multiple functional states of the channel. Do the authors identify similar states of the TRPV2 channel by activation via C16? Active /inactive modus of the channel? In addition, the patch clamp experiments illustrated in figure 2a and figure 3c showed an activation of the channel by C16 application followed by a reduction of the current amplitude in the presence of C16. How do the authors explain the transient effect of C16 in patch clamp experiments? Could the transient effect be linked to the different states of the channel? Is the transient response to C16 calcium dependent?
- The authors showed single channel activity data are as a single representative trace. However, to provide convincing evidence the data presentation should be more elaborated. Data should be shown in basal conditions before application of C16 and after C16 application for multiple cells. A more detailed analysis is required to come up with convincing experiments. Did the authors investigate the different mode of applications of C16 to gather additional information on the mode of action (inside/outside). Is the channel conductance similar for C16 as for THC and 2-APB? What about the co-application of probenecid and C16. Is the effect of RR shown in panel c reversible?
- The authors claim a role for probenecid as channel agonist of TRPV2. However, direct application of probenecid showed a limited increase in intracellular calcium concentration at a very high dose (10-100 mM), at concentrations that are even higher to activate TRPV1 and TRPA1. Also the cryo-EM experiments don't show clear evidence for a direct bound of probenecid and TRPV2 (neither the probenecid alone or the combination of C16 and probenecid). The authors have to show whether probenecid is a real agonist of TRPV2 (eg. via patch clamp experiments, single channel recordings) or rather a modulator of the channel activity. Is the potentiation of probenecid C16 dependent? For some

TRP channels like TRPM3, it is described that some compounds at low concentrations act as a modulator of the channel activity and only at very high doses could increase channel activity by its own. Interestingly, co-application of clotrimazole and the neurosteroid PS induces strong potentiation of the inward current of TRPM3 by opening of an alternative ion permeation pathway. Could the authors exclude a similar mechanism for TRPV2 and probenecid? (Vriens et al 2014, Nature Chem Biol).

Minor:

- As the authors stated in the introduction, no evidence is available to claim that TRPV2 is a thermoreceptor. Please erase the name of thermoreceptor in the entire manuscript.
- - The authors have to show the synergism between probenecid on C16. Currently, no comparative studies are included to indicate the synergism of probenecid (C16, probenecid, C16+probenecid). Is the synergistic effect of probenecid dependent of C16 or could probenecid also potentiate CBD or 2-APB responses in TRPV2?
- Please discuss the difference in potency between Δ^9 -THC and C16. How do the authors explain the difference in potency based on the current models.
- The authors use an extremely high dose of RR (40 μ M and 100 μ M) to block the synergistic effect of C16 and probenecid? At this high concentrations of RR all channels in the PM will be blocked, and therefore make the experiments inconclusive. Was the effect of RR shown in the single cell data (figure 4 panel c) reversible? It is advised to use more selective TRPV2 inhibitors as was recently by Bluhm et al 2022, or use lower doses of RR.
- Please discuss the expression levels of TRPV1 and TRPA1 in RBL-2H3 cells and HMC-1.2 cells. Since high dose of RR (10 μ M) could also inhibit other ion channels
- Suggestion to visualize the PC-displacement-activation-mechanism by the use of cartoons?
- Typo P7 line 249 TPRV2 \diamond TRPV2
- Typo P9 line 354 TPRV2 \diamond TRPV2

We thank the reviewers for the evaluation and for the helpful suggestions and comments to improve the manuscript. We have now addressed all comments and amended the manuscript as outlined below, also providing considerable amounts of additional data. Remarks and questions from the reviewers are shown in black. Our responses are shown in blue, and figures included only in this point-to-point answer can be found at the end of the document and are referred to as Figures A1 and A2.

REVIEWER COMMENTS

Reviewer #1 (Remarks to the Author):

Here the authors study the opening of TRPV2 by a cannabinoid compound, C16, and the TRPV2 agonist probenecid. The first part of the study focuses on the functional characterization of TRPV2 activation by the two ligands. The authors find that the effects of C16 and probenecid are synergistic, i.e. that application of one followed by application of the other produces a larger current than application of either one on its own. The authors surmise that the ligands have different binding sites and mechanisms of action.

The second part of the study aims to identify the ligand binding sites and their mechanisms of action via analysis of cryo-EM structures of rat TRPV2 in the presence of C16 and probenecid.

It is my opinion that there would be some interest in the data amongst the TRP channel community. However, there are several issues that need to be addressed.

Authors: We are thankful to the reviewer for the valuable and constructive criticism on our study. We are pleased to know that the reviewer finds the study of interest, and trust that we have satisfactorily addressed below all points raised by the reviewer.

Major concerns:

1. I would like to see some more evidence that the C16/probenecid effect isn't due to use dependence in TRPV2. For example, how does TRPV2 respond to repeated exposures to C16 and probenecid, respectively? Does the EC50 for each change upon repeated exposure? And how do those values compare to the change in TRPV2 sensitivity to probenecid upon C16 exposure? Does C16 need to be present to see the increased effect of probenecid, and vice versa? Or, can the effect be seen after mere exposure to C16 / probenecid (i.e., one ligand is washed out before application of the second one)?

Please see the section on ligand cross-sensitization in Liu, Beiyong, and Feng Qin. "Use Dependence of Heat Sensitivity of Vanilloid Receptor TRPV2." *Biophysical journal* vol. 110,7 (2016): 1523-1537. doi:10.1016/j.bpj.2016.03.005.

Please also see the study of the related TRPV3 channel: Liu B, Yao J, Zhu MX, Qin F. Hysteresis of gating underlines sensitization of TRPV3 channels. *J Gen Physiol*. 2011;138(5):509-520. doi:10.1085/jgp.201110689.

Authors: The use dependence for heat and ligand agonist activation of TRPV2 is indeed an intriguing phenomenon. Whereas the use dependence for heat responses is consistently reported in the literature and observed both in the whole-cell and isolated patch configurations[1], ligand use dependence is less clear because when 2-APB was applied repeatedly at the same concentration, no use dependence was observed in whole-cell recordings [2] and even not always in inside-out membrane patches[1]. Both studies were using rat TRPV2 expressed in HEK293 cells.

Nevertheless, to address the possibility of ligand use-dependent activation of TRPV2 and other concerns raised by the reviewer, we have added a collection of new data from patch-clamp experiments in the whole-cell configuration demonstrating that repeated exposure to neither C16 nor probenecid displayed obvious use dependence that could explain the synergy between probenecid and C16 under the same experimental conditions (Fig. 2a, b and Supplementary Figs. 4, 5 and 6). Furthermore, the heat response in the presence of probenecid after washout of C16 returned to a magnitude similar to that before co-application of C16 (Supplementary Fig. 6b). Moreover, no effect of C16 at a subliminal concentration (10 μ M), acting in synergy with probenecid, was observed in the absence of probenecid (Fig. 2a). Also, the response to a second application of probenecid in the presence of C16 vehicle was not different from the initial probenecid response (Fig 2c, d). Thus, both compounds need to be present for a synergistic non-use-dependent effect on TRPV2.

2. Ligand binding sites need to be validated by electrophysiology. This is especially important because ligands are small, the densities shown in the figures are weak, and cryo-EM does not offer a way to unambiguously identify bound molecules. Also, the authors should show half maps for the ligand densities.

Authors: We have performed whole-cell patch-clamp studies of 5 mutants (D536E, F540A, S526A, S526E and L636A) to shed light on C16 interaction with TRPV2 at D536, F540, S526 and L636, all of which are critical residues around the proposed binding site. These data are included in our updated manuscript (Fig. 6 and Supplementary Figs. 15 and 16). The data support that D536 is a key ligand for C16, and of importance for a proper synergistic interaction between the C16 binding site and a separate binding site for probenecid. An inhibitory effect of C16 was revealed by the S526E and L636A mutants, indicating that targeting the C16 binding site may be a suitable approach to fine tune TRPV2 activity. The inability of the non-functional F540A mutant to respond to ligands (C16 and probenecid), heat or voltage (not shown) further support that C16 is interacting with important TRPV2 parts within the vanilloid pocket, which is a nexus for integrating various chemical and temperature stimuli in TRPV channels. The half maps of the two C16 binding modes are shown in Supplementary Fig. 14. As for the weak cryo-EM densities detected for C16, they may well relate to the relatively low resolution of the structures. While there is some ambiguity with regard to the sidechains of F540 and Y525, together with the electrophysiological data and our mutagenesis studies, we are confident that C16 is bound at the indicated position. Nevertheless, the combined electrophysiological and cryo-EM data point towards C16 being bound at the pinpointed site adjacent to Y525, S526, D536, F540 and L636. Of note, the mutant Y525A was excluded in our mutagenesis studies as it displays a global loss-of-function phenotype[3].

3. The same applies to the proposed mechanisms – these should be tested by mutagenesis and electrophysiology.

Authors: The reviewer raises an important point, which, as outlined in the manuscript, has already been supported by available data from several groups with regard to the proposed mechanism. Most notably, the “lipid-displacement-activation/inhibition-mechanism” is substantiated by several studies that have shown that chemical stimuli elicit TRPV-channel activation/inhibition by promoting the release of bioactive lipids from critical allosteric regulatory sites such as the vanilloid pocket[4-11]. This takes place via activators in TRPV1-4, while inhibitors achieve a similar effect in TRPV5-6. Specifically, in TRPV1, the phosphate head group of the lipid in the vanilloid pocket is stabilized by polar interactions and salt-bridges with R557 or K571[6]. The equivalent residues in TRPV2 are R517 and K531, and they also interact with the lipid head in our structure as shown in Figure 4e. In addition, R470, Q483, K484 and Q596 of TRPV6 have been reported to stabilize the head group of the lipid present in the vanilloid pocket, as we describe for the equivalent lipid surrounding residues R517, N530, K531 and Q663 of TRPV2 in Figure 4e. The vital role of R470 for lipid binding in TRPV6 has been validated using both structural and functional studies of the R470E form of TRPV6[12].

Along the same line, we propose that in our C16-bound TRPV2 structure, lipid in the vanilloid pocket is displaced by a non-protein density (tentatively assigned as CHS). This is also observed in TRPV2 structures treated with RTX, displaying exchange of the lipid in the vanilloid pocket for RTX. Based on the structural and electrophysiological studies, it was suggested that a hydrogen bond triad is an essential part of the transition between closed and open states of the channel[13, 14]. For TRPV1 and TRPV2, the channel activity is negatively affected in mutants losing this hydrogen bond triad[13, 14]. In our manuscript, we proposed a homologous mechanism, with the hydrogen triad formed by Y544, T604 and Y629 serving a vital step for widening of the LG upon C16 treatment.

4. The authors seem to assign a site to CHS without any supporting evidence. The site is used in downstream mechanistic explanations. I would strongly urge the authors to tone this down, given that no evidence is provided for CHS binding in that position.

Authors: We agree with the reviewer that the CHS site cannot be assigned unambiguously. In this light, we have toned down the claims regarding CHS in the manuscript.

5. The structural alignments are confusing – it is difficult to discern movements of individual domains when no fixed point exists in the alignment. I would suggest remaking the figure 7 a-c with a better alignment of the two structures and re-evaluating the conformational changes.

Authors: We respect the suggestions of reviewer, yet we would like to keep our structural alignment, comparing the global changes of the proteins, also considering that the proposed C16 binding site is located at the interface between the monomers, while the ion conductance pore-width remains essential. Moreover, in our view, there is generally a risk for overinterpretation of the data with monomer-to-monomer alignments only (in particular for

regions distant to the selected alignment region). Nonetheless, we here provide the requested analysis, using alignments of transmembrane segments S1-S4 of separate monomers (Fig. A2). Notably, the trends of the structural changes are maintained independent of the employed alignment region, further supporting that the original figures can be kept.

6. The structural analysis lacks some quantification – RMSD values, angles, distances, etc.

Authors: The RMSD values were indicated in Supplementary Table 1 already in the first submitted version. In relation to the previous remark, Supplementary Table 1 includes RMSD values for global alignments, and separate alignments using either S1-S6 or S1-S4 comparisons. This way, we hope to further illustrate which regions that are shifting. Regarding angles and distances, we agree that some of the previous figures were short of information. Consequently, we have now included additional values to quantify the changes between structures in the figures.

7. C16 alone barely activates TRPV2 in HEK cells but in bilayers it causes a 0.5 open probability? How is this explained? How does probenecid + C16 affect P_o in bilayers? Also, even though data for the C16 effect on empty bilayers has been published before, this should be shown here too. It's an important control that needs to be presented.

Authors: One explanation could be that it is easier for C16 to access its binding site in the bilayer recordings and that additional mechanisms in the intact cell modify the C16 evoked TRPV2 activity at the single-channel level affecting the open probability and/or conductance. However, we have added a collection of new data from patch-clamp experiments in the whole-cell configuration clearly demonstrating activation by C16 and probenecid alone as well as the synergy between probenecid and C16 (Fig. 2a, b and e and Supplementary Figs. 4, 5 and 6). Regarding lipid bilayer electrophysiology, these experiments were performed to confirm a direct interaction between C16 and TRPV2 before taking on cryo-EM studies. We performed experiments with purified rat TRPV2 reconstituted into artificial lipid bilayers for electrophysiological recordings with focus on C16, as probenecid has already been shown to activate purified rat TRPV2 when reconstituted into lipid bilayers[15]. Nevertheless, in few experiments ($n = 2$), it was confirmed that probenecid at 100 μM alone and combined with C16 activates purified rat TRPV2 when reconstituted into lipid bilayers (Supplementary Fig. 7). We have also included basal conditions before the application of C16 and probenecid, and their lack of effect in bilayers without TRPV2 (Supplementary Fig. 7). These data are in perfect agreement with results from the whole-cell recordings of heterologously expressed TRPV2 presented in this study.

8. The choice to analyze the pore opening by removing side chains is curious. The C α distances aren't necessarily a good measure of the extent of pore opening, so the argument that something is more or less open based on HOLE profiles calculated from polyaniline models is perhaps a little misleading. I would argue that the language here should not include terms "open" or "closed" but instead refer to C α distances, with a clear acknowledgement that side chains will contribute to the functional state of the pore. Also, given the high resolution of the structures obtained here, the figures should also include a

HOLE profile of the structures with side chains so that there is no doubt about what the actual data tells us.

Authors: We kindly disagree with the reviewer regarding this point. As can be observed from Supplementary Figs. 20 it can be difficult to properly assign the sidechains at the intermediate resolutions frequently obtained for cryo-EM. This is also true for the pore-lining transmembrane segment S6, which (together with S5 and PH) often displays relatively low resolution compared to the other parts (S1-S4) of the TM-domains of TRP-channel structures. As a side note, this difficulty also applies to many already reported structures of TRP channels (for which pore width calculations are provided despite moderate resolution). Consequently, pore width calculations based on the sidechain positions can be model-biased and hence overinterpreted, whereas the calculations based on polyaniline chains accounts for these types of errors. Thus, as stated in the manuscript, we prefer the more unbiased comparisons of Calpha distances. This however does not mean that we imply that sidechains are not important for establishing the pore width, and of course sidechains will contribute to the functional state of the pore as indicated by the reviewer, but for comparisons of structures at an intermediate resolution (around 3.5 Å), we are rather certain that the exploited Calpha comparisons are more robust and unbiased. Nevertheless, we have generated the requested pore-width analysis using sidechains (Fig. A1). We observe that the LG of all our here presented ligand treated structures is wider than the LG of the available apo structures of TRPV2 (PDB-ID 6U84, overall resolution 3.7 Å: LG=0.70 Å; 6U86, 4.0 Å: LG=1.2 Å). Finally, we agree that we should use more exquisite terms than only “open” and “closed” to appropriately describe the structures. This has now been accounted for in the revised version of the manuscript.

9. The reconstructions all reach good overall resolutions. However, the local resolution plots are missing and should be included.

Authors: We agree with reviewer. The local resolution plot for each structure is now included in Supplementary Fig. 20.

Reviewer #2 (Remarks to the Author):

The manuscript untitled: 'Cannabinoid non-cannabidiol site modulation of TRPV2 structure and function' submitted by Zhang et al, identified tetrahydrocannabinol as an agonist of TRPV2. Via cryo-EM studies the authors identified the interaction site of C16 on TRPV2, a site that was similar for the previously described ligand pocket for CBD and 2-APB. In addition, the authors describe a synergistic effect of C16 and probenecid in mouse and human TRPV2 and showed that C16 and probenecid have different structural effects on the channel pore.

The presented work is a nice follow up study of recently published SAR studies work on TRPV2 of Pumroy et al 2022 in which a ligand interaction site for TRPV2 was identified. However, the current study has a lack on mutagenesis as a proof-of-concept. This is important to validate the conclusions, in particular the part in which the interaction of TRPV2 with probenecid is described.

Authors: We are thankful to the reviewer for the very valuable and constructive criticism on our study. We are pleased to know that the reviewer finds the study of interest, and trust that we have satisfactorily addressed below all points raised by the reviewer.

Major comments:

- Despite the many benefits of the cryo-EM technique and the availability of the protein structure, the proposed method also has its limitations, as the purified proteins are not located in the PM of the cell and are crystalized without any physiological membrane potential. This is an important factor as all TRP channels (including TRPV2) are all voltage dependent. The conclusions of the manuscript are based on comparison studies of new and previously published cryo-EM models. However, the novel findings should be supported by mutagenesis studies as a proof-of-concept. For example: (1) No strong evidence is provided that the vanilloid pocket region is occupied with lipids like phosphocholine or PI. This is based on earlier speculations, but no hard evidence is proven. The authors should provide additional evidence via mutagenesis studies and neutralization of the positive charges at position R517 and K531. (2) the identification of the C16 binding site. The authors identified a potential region where the C16 could interact with the channel. However, to further provide stronger evidence mutagenesis studies and functional read-outs are required in which a series of key residues (H521 or Y525) should be neutralized as a proof of concept.

Authors: We have performed whole-cell patch-clamp studies of 5 mutants (D536E, F540A, S526A, S526E and L636A) to shed light on C16 interaction with TRPV2 at D536, F540, S526 and L636, all of which are critical residues around the proposed binding site. These data are included in our updated manuscript (Fig. 6 and Supplementary Figs. 15 and 16). The data support that D536 is a key ligand for C16, and of importance for a proper synergistic interaction between the C16 binding site and a separate binding site for probenecid. An inhibitory effect of C16 was revealed by the S526E and L636A mutants, indicating that targeting the C16 binding site may be a suitable approach to fine tune TRPV2 activity. The inability of the non-functional F540A mutant to respond to ligands (C16 and probenecid), heat or voltage further support that C16 is interacting with important TRPV2 structures within the vanilloid pocket, which is a nexus for integrating various chemical and temperature stimuli in

TRPV channels. The mutant Y525A was excluded in our mutagenesis studies as it displays a global loss-of-function phenotype[3].

As also indicated to question 3 from Reviewer #1, our conclusions are also supported by reminiscent results for other TRPV channels. In a study of TRPV6, the equivalent of R517 (R470 in TRPV6) was substituted with a glutamate, which resulted in a smaller size lipid located to the vanilloid pocket and closed configuration of the channel. Similarly, the R557E form of TRPV1 (targeting the homologous residue as R517 in TRPV2) is associated with a complete loss of activation by capsaicin. Thus, it must be considered well-established that the equivalent residue of R517 in TRPV2 can modulate the binding to the vanilloid pocket in TRPV channels, and hence also the channel conductance. At the same time, the pocket and even the R517 position is sensitive for mutations, as R557 represents a dead mutant of TRPV1, in agreement with our new mutagenesis data.

- In a recent study of Pumroy et al, 2021 Nature Communications evidence was provided for a multiple functional states of the channel. Do the authors identify similar states of the TRPV2 channel by activation via C16? Active /inactive modus of the channel? In addition, the patch clamp experiments illustrated in figure 2a and figure 3c showed an activation of the channel by C16 application followed by a reduction of the current amplitude in the presence of C16. How do the authors explain the transient effect of C16 in patch clamp experiments? Could the transient effect be linked to the different states of the channel? Is the transient response to C16 calcium dependent?

Authors: The effect of probenecid was transient in whole-cell recordings when Ca^{2+} was present in the extracellular solution (Fig. 2c and Fig. 3c). Likewise, in the study by Bang and colleagues[16], probenecid but not 2-APB evoked a transient current response in whole-cell patch-clamp recordings with Ca^{2+} in the extracellular bath solution. Together, this could indicate that the transient response to probenecid is related to the presence of extracellular Ca^{2+} and involves an activation mechanism distinct from that of 2-APB. However, transient responses are also observed for 2-APB and CBD at higher concentrations, visible just before washout, in Ca^{2+} -free recording solutions[3]. Perhaps, the strength and duration of the stimulus also contribute to the transient responses, possibly reflecting different conformational states of the channel? We also note that the two separate structures in the presence of C16 may be interpreted as separate states (TRPV2_{C16-2} more open than TRPV2_{C16-1}), as stated in the manuscript.

- The authors showed single channel activity data are as a single representative trace. However, to provide convincing evidence the data presentation should be more elaborated. Data should be shown in basal conditions before application of C16 and after C16 application for multiple cells. A more detailed analysis is required to come up with convincing experiments. Did the authors investigate the different mode of applications of C16 to gather additional information on the mode of action (inside/outside). Is the channel conductance similar for C16 as for THC and 2-APB? What about the co-application of probenecid and C16. Is the effect of RR shown in panel c reversible?

Authors: These experiments were performed to confirm a direct interaction between C16 and TRPV2 before taking on cryo-EM studies. We performed experiments with purified rat TRPV2

reconstituted into artificial lipid bilayers for electrophysiological recordings with focus on C16, as probenecid has already been shown to activate purified rat TRPV2 when reconstituted into lipid bilayers[15]. As shown in Supplementary Fig. 7, we have included basal conditions before the application of C16 and probenecid, and their lack of effect in bilayers without TRPV2. The high purity of TRPV2 and its cryo-EM structures leave no doubt that the recorded single-channel activity is mediated by TRPV2 with proper structure and function (Supplementary Fig. 22). Not all cryo-EM studies on TRP channels include bilayer patch-clamp single-channel recordings of the purified channel[3], most likely because it involves critical and time-consuming steps apart from expression/purification such as channel incorporation into the lipid bilayer and tedious recordings and analysis of single-channel activity. In fact, one key limitation for us is the channel incorporation into the lipid bilayer. Thus, although it would be interesting to collect further detailed information on the mode of action and additional ligands, this has to be investigated in studies separate from the present one. Nevertheless, based on our own experience of bilayer recordings with purified human TRPA1[17-21], we are confident here to provide enough key bilayer recordings in support of a direct action of C16 on rat TRPV2.

- The authors claim a role for probenecid as channel agonist of TRPV2. However, direct application of probenecid showed a limited increase in intracellular calcium concentration at a very high dose (10-100 mM), at concentrations that are even higher to activate TRPV1 and TRPA1. Also the cryo-EM experiments don't show clear evidence for a direct bound of probenecid and TRPV2 (neither the probenecid alone or the combination of C16 and probenecid). The authors have to show whether probenecid is a real agonist of TRPV2 (eg. via patch clamp experiments, single channel recordings) or rather a modulator of the channel activity. Is the potentiation of probenecid C16 dependent? For some TRP channels like TRPM3, it is described that some compounds at low concentrations act as a modulator of the channel activity and only at very high doses could increase channel activity by its own. Interestingly, co-application of clotrimazole and the neurosteroid PS induces strong potentiation of the inward current of TRPM3 by opening of an alternative ion permeation pathway. Could the authors exclude a similar mechanism for TRPV2 and probenecid? (Vriens et al 2014, Nature Chem Biol).

Authors: Probenecid was first shown to activate TRPV2 by Bang and colleagues[16] and is generally claimed to be an agonist of TRPV2[15, 22, 23]. As shown in Supplementary Fig. 2, probenecid causes substantial concentration-dependent Ca^{2+} responses between **1 - 10 mM**, which fits with a role of probenecid as an agonist of TRPV2. Furthermore, as shown in whole-cell recordings, probenecid within the same concentration interval produces concentration-dependent TRPV2 currents also adding evidence of an agonistic action of probenecid on TRPV2 (Supplementary Fig. 5). As originally demonstrated by Hyunh and colleagues[15] and confirmed here (Supplementary Fig. 7), probenecid at 100 μM can directly activate TRPV2 when as a purified protein it is reconstituted into artificial lipid bilayers. Our cryo-EM structure in the presence of probenecid also suggests that probenecid serves as an activator, as the structure is more open than closed TRPV2. Furthermore, in whole-cell patch-clamp experiments, we show that the potentiation of probenecid is dependent on C16 and vice versa (Fig. 2 and Supplementary Fig. 6). It may very well be that probenecid is a modulator at low concentrations and activator at high concentrations, possibly leading to desensitization that could be beneficial in a physiological relevant context as discussed for some TRPA1

modulators/activators[24, 25]. At this stage, we cannot exclude that the synergistic effect of probenecid and C16 involves an alternative ion permeation pathway as shown for TRPM3 with clotrimazole and neurosteroid PS. However, from a structural point of view we do not observe any indications of such alternative pathways.

Minor:

- As the authors stated in the introduction, no evidence is available to claim that TRPV2 is a thermoreceptor. Please erase the name of thermoreceptor in the entire manuscript.

Authors: We agree with the reviewer. 'Thermoreceptor' is now only used for relevant TRP channels in the manuscript.

- - The authors have to show the synergism between probenecid on C16. Currently, no comparative studies are included to indicate the synergism of probenecid (C16, probenecid, C16+probenecid). Is the synergistic effect of probenecid dependent of C16 or could probenecid also potentiate CBD or 2-APB responses in TRPV2?

Authors: In addition to the data generated by Ca²⁺-imaging and whole-cell patch-clamp electrophysiology on HEK293 and RBL-2H3 cells (Figs 1, 2c,d,f and 3), data that were presented in the original submitted manuscript, we have added a collection of new data from patch-clamp experiments in the whole-cell configuration clearly demonstrating synergy between probenecid and C16 (Fig. 2a, b and e and Supplementary Figs. 4, 5 and 6). We also refer to the study by Bluhm and colleagues[23], demonstrating a similar synergistic effect on rat TRPV2 of probenecid when combined with 2-APB. A synergistic effect of 2-APB and CBD on rat TRPV2 has also been demonstrated recently by Pumroy and colleagues[3], and it will be interesting to investigate the many possible combinations of ligands such as probenecid, C16, CBD, 2-APB and oxidants. However, such studies in combination with cryo-EM are unfortunately extensive and thus beyond the scope of the present study.

- Please discuss the difference in potency between Δ^9 -THC and C16. How do the authors explain the difference in potency based on the current models.

Authors: Because the use of Δ^9 -THC as a research tool has become even more strictly regulated than previously[26], we have chosen not to include this psychotropic compound in the present study. However, a structure-activity study, under identical experimental conditions, with Δ^9 -THC and other derivatives thereof with stepwise truncation of its alkyl sidechain including C16[26] on TRPV2 would indeed be very interesting and hopefully a feasible project in the future. At the moment, we can only compare the potency of C16 with that of Δ^9 -THC already reported in the literature[27], also measuring changes in intracellular Ca²⁺ as readout for rat TRPV2 activation by Δ^9 -THC, albeit with a different Ca²⁺ dye as in our experiments. Neepner and colleagues report an EC₅₀ of 16 μ M for Δ^9 -THC, which makes Δ^9 -THC approximately 6 times more potent than C16 as an agonist for rat TRPV2. Also, when compared with Δ^9 -THC, cannabidiol with an EC₅₀ of 3.7 μ M is approximately 5 times more potent on rat TRPV2 using the same experimental conditions[27, 28]. In contrast to cannabidiol, classical cannabinoids such as Δ^9 -THC and C16 are ABC-tricyclic terpenoid compounds with a benzopyran moiety (see Supplementary Fig. 8). Thus, it seems likely that

Δ^9 -THC interacts with the C16 binding site rather than the cannabidiol site[3]. However, further studies as discussed above are necessary to conclude that this is indeed the case.

- The authors use an extremely high dose of RR (40 μ M and 100 μ M) to block the synergistic effect of C16 and probenecid? At this high concentrations of RR all channels in the PM will be blocked, and therefore make the experiments inconclusive. Was the effect of RR shown in the single cell data (figure 4 panel c) reversible? It is advised to use more selective TRPV2 inhibitors as was recently by proposed by Bluhm et al 2022, or use lower doses of RR.

Authors: With the exception of the experiments performed on RBL-2H3 cells in Fig. 3b and in bilayer recordings (now Supplementary Fig. 7c) we have used a RR concentration of 10 μ M to block TRPV2[29], although a 10 times higher concentration was needed to completely block cannabinoid[27, 28] and 2-APB[2] activation of rat TRPV2 and mouse TRPV2, respectively. Thus, it is not inappropriate to use 100 μ M RR to secure complete inhibition of TRPV2, as was also shown for heat by Caterina and colleagues[29]. Nevertheless, 10 μ M RR completely inhibited the effect of C16 and probenecid combined in RBL-2H3 cells when also measured by patch-clamp electrophysiology (Fig. 3c), and in HEK293 cells expressing rat TRPV2 (Fig. 2f), data that were included in the original submission. The reversibility of the RR inhibitory effect on purified rat TRPV2 activity when reconstituted into planar lipid bilayers was not examined. Our focus was on providing evidence that C16 directly activates TRPV2, since probenecid has already been shown to activate purified rat TRPV2 reconstituted into artificial lipid bilayers[15]. Nevertheless, we have added few experiments confirming that probenecid alone and combined with C16 directly activates purified rat TRPV2 (Supplementary Fig. 7). Among the various TRP channels inhibited by RR, TRPV2 and TRPA1 seem to need somewhat higher concentrations of RR for channel inhibition (see IUPHAR/BPS guide to pharmacology; <https://tinyurl.se/5Am>). It is unfortunate that more selective TRPV2 inhibitors were not available during the work on this study. However, although valdecoxib selectively inhibited TRPV2 relative TRPV1, TRPV3 and TRPV4[23], still a high concentration (100 μ M) is needed for proper inhibition of TRPV2 currents in both HEK293 and RBL-2H3 cells[23] as also shown in HEK293T cells (Fig. 2e).

- Please discuss the expression levels of TRPV1 and TRPA1 in RBL-2H3 cells and HMC-1.2 cells. Since high dose of RR (10 μ M) could also inhibit other ion channels

Authors: With the knowledge that C16 and probenecid also activate TRPA1 as well as TRPV1, which we show here for the first time, we had to consider TRPA1 and TRPV1 as possible targets when studying the effect of these compounds on TRPV2 mast cell function. The expression of TRPV1 compared to TRPA1 seems more robust and convincing in both RBL-2H3 and HMC-1.2 cells[30, 31], and both channels are blocked by RR at the concentrations used here (10 and 100 μ M). Therefore, we performed functional experiments with both antagonists and agonists of TRPV1 and TRPA1 to rule out that the effects of C16 and probenecid involved these TRP channels (Fig. 3b, e).

Notably, RR can block a number of TRPV channels, of which TRPV3 and TRPV5 have not been reported in mast cells, as well as other targets that are present in mast cells and involved in intracellular Ca^{2+} homeostasis[30] (IUPHAR/BPS guide to pharmacology; <https://tinyurl.se/5Am>). Regarding the use of valdecoxib that did not inhibit TRPV1, TRPV3

and TRPV4[23], a high concentration (100 μ M) is needed for proper inhibition of TRPV2 responses[23] (see also new experiments in Fig 2e), and off-target effects cannot be excluded for valdecoxib, as shown for the closely related celecoxib and parecoxib interacting with e.g., K^+ and Ca^{2+} channels[32, 33], when used in more complex cellular assays such as mast cells. Nevertheless, in our mast cell experiments C16 and probenecid alone and combined display the same pharmacological profile as in all other functional experiments where the expression of TRPV2 is induced. Thus, we are confident to suggest TRPV2 represents an interesting target for further studies of mast cell function including the use of probenecid and C16 combined.

- Suggestion to visualize the PC-displacement-activation-mechanism by the use of cartoons?

Authors: Our ambition is that modified Fig. 10 hopefully illustrates the overall mechanism involved in ligand exchange.

- Typo P7 line 249 TPRV2 \rightarrow TRPV2

Authors: Corrected.

- Typo P9 line 354 TPRV2 \rightarrow TRPV2

Authors: Corrected.

Figure A1. Pore calculations with side chains being included for each ligand-treated TRPV2 structure, TRPV2_{C16-1}, TRPV2_{C16-2} and TRPV2_{probenecid}, respectively, as well as for the available apo TRPV2 structures TRPV2_{apo-1} and TRPV2_{apo-2}. The overall resolution of the structures is indicated in the title above each analysis. Results of the structures for the combinations of ligands are not presented here due to rather poor local resolution at the LG, see Supplementary Figure 20.

Figure A2. C16-induced conformational changes. Overlay of the TRPV2_{C16-2} (dark green) and TRPV2_{apo-1} (PDB-ID: 6U84, pink) and TRPV2_{C16-2} (dark green) and TRPV2_{PLG} (PDB-ID: 6WKN, purple) structures based on one fixed monomer except a, d, which are aligned on tetramer. a-c Membrane views of TRPV2_{C16-2} and TRPV2_{apo-1} showing (a) two opposite S5-PH-S6 units, (b) a single subunit (S1-S5) and an adjacent subunit (showing S5-PH-S6-TRP helix), and (c) the local changes around the suggested C16 binding site. d-e Membrane views similar to a-c comparing TRPV2_{C16-2} and TRPV2_{PLG}. Arrows indicate movements of TRPV2_{C16-2} relative to TRPV2_{apo-1} and TRPV2_{PLG}, respectively. Local conformational changes are highlighted with stars. C16 is displayed as pink sticks in b, c and e, f.

References

1. Liu, B. and F. Qin, *Use Dependence of Heat Sensitivity of Vanilloid Receptor TRPV2*. *Biophys J*, 2016. **110**(7): p. 1523-1537.
2. Juvin, V., et al., *Pharmacological characterization and molecular determinants of the activation of transient receptor potential V2 channel orthologs by 2-aminoethoxydiphenyl borate*. *Mol Pharmacol*, 2007. **72**(5): p. 1258-68.
3. Pumroy, R.A., et al., *Structural insights into TRPV2 activation by small molecules*. *Nat Commun*, 2022. **13**(1): p. 2334.
4. Singh, A.K., et al., *Structural bases of TRP channel TRPV6 allosteric modulation by 2-APB*. *Nat Commun*, 2018. **9**(1): p. 2465.
5. Pumroy, R.A., et al., *Molecular mechanism of TRPV2 channel modulation by cannabidiol*. *Elife*, 2019. **8**.
6. Gao, Y., et al., *TRPV1 structures in nanodiscs reveal mechanisms of ligand and lipid action*. *Nature*, 2016. **534**(7607): p. 347-51.
7. McGoldrick, L.L., et al., *Opening of the human epithelial calcium channel TRPV6*. *Nature*, 2018. **553**(7687): p. 233-237.
8. Hughes, T.E.T., et al., *Structural insights on TRPV5 gating by endogenous modulators*. *Nat Commun*, 2018. **9**(1): p. 4198.
9. Dang, S., et al., *Structural insight into TRPV5 channel function and modulation*. *Proc Natl Acad Sci U S A*, 2019. **116**(18): p. 8869-8878.
10. Shimada, H., et al., *The structure of lipid nanodisc-reconstituted TRPV3 reveals the gating mechanism*. *Nat Struct Mol Biol*, 2020. **27**(7): p. 645-652.
11. Deng, Z., et al., *Gating of human TRPV3 in a lipid bilayer*. *Nat Struct Mol Biol*, 2020. **27**(7): p. 635-644.
12. Yelshanskaya, M.V., et al., *Structure and function of the calcium-selective TRP channel TRPV6*. *J Physiol*, 2021. **599**(10): p. 2673-2697.
13. Zubcevic, L., et al., *Conformational plasticity in the selectivity filter of the TRPV2 ion channel*. *Nat Struct Mol Biol*, 2018. **25**(5): p. 405-415.
14. Zubcevic, L., et al., *Symmetry transitions during gating of the TRPV2 ion channel in lipid membranes*. *Elife*, 2019. **8**.
15. Huynh, K.W., et al., *Structural insight into the assembly of TRPV channels*. *Structure*, 2014. **22**(2): p. 260-8.
16. Bang, S., et al., *Transient receptor potential V2 expressed in sensory neurons is activated by probenecid*. *Neurosci Lett*, 2007. **425**(2): p. 120-5.
17. Moparthi, L., et al., *Human TRPA1 is a heat sensor displaying intrinsic U-shaped thermosensitivity*. *Sci Rep*, 2016. **6**: p. 28763.
18. Moparthi, L., et al., *Calcium activates purified human TRPA1 with and without its N-terminal ankyrin repeat domain in the absence of calmodulin*. *Cell Calcium*, 2020. **90**: p. 102228.
19. Moparthi, L., et al., *Human TRPA1 is intrinsically cold- and chemosensitive with and without its N-terminal ankyrin repeat domain*. *Proc Natl Acad Sci U S A*, 2014. **111**(47): p. 16901-6.
20. Moparthi, L. and P.M. Zygmunt, *Human TRPA1 is an inherently mechanosensitive bilayer-gated ion channel*. *Cell Calcium*, 2020. **91**: p. 102255.

21. Moparthy, L., et al., *The human TRPA1 intrinsic cold and heat sensitivity involves separate channel structures beyond the N-ARD domain*. Nat Commun, 2022. **13**(1): p. 6113.
22. Solis-Lopez, A., et al., *Analysis of TRPV channel activation by stimulation of FCepsilonRI and MRGPR receptors in mouse peritoneal mast cells*. PLoS One, 2017. **12**(2): p. e0171366.
23. Bluhm, Y., et al., *Valdecoxib blocks rat TRPV2 channels*. Eur J Pharmacol, 2022. **915**: p. 174702.
24. Zygmunt, P.M. and E.D. Högestätt, *Trpa1*. Handb Exp Pharmacol, 2014. **222**: p. 583-630.
25. Talavera, K., et al., *Mammalian Transient Receptor Potential TRPA1 Channels: From Structure to Disease*. Physiol Rev, 2020. **100**(2): p. 725-803.
26. Andersson, D.A., et al., *TRPA1 mediates spinal antinociception induced by acetaminophen and the cannabinoid Delta(9)-tetrahydrocannabinol*. Nat Commun, 2011. **2**: p. 551.
27. Nepper, M.P., et al., *Activation properties of heterologously expressed mammalian TRPV2: evidence for species dependence*. J Biol Chem, 2007. **282**(21): p. 15894-902.
28. Qin, N., et al., *TRPV2 is activated by cannabidiol and mediates CGRP release in cultured rat dorsal root ganglion neurons*. J Neurosci, 2008. **28**(24): p. 6231-8.
29. Caterina, M.J., et al., *A capsaicin-receptor homologue with a high threshold for noxious heat*. Nature, 1999. **398**(6726): p. 436-41.
30. Freichel, M., J. Almering, and V. Tsvilovskyy, *The Role of TRP Proteins in Mast Cells*. Front Immunol, 2012. **3**: p. 150.
31. Naert, R., A. Lopez-Requena, and K. Talavera, *TRPA1 Expression and Pathophysiology in Immune Cells*. Int J Mol Sci, 2021. **22**(21).
32. Frolov, R.V. and S. Singh, *Evidence of more ion channels inhibited by celecoxib: KV1.3 and L-type Ca(2+) channels*. BMC Res Notes, 2015. **8**: p. 62.
33. Liu, Y.Y., et al., *Parecoxib, a selective blocker of cyclooxygenase-2, directly inhibits neuronal delayed-rectifier K(+) current, M-type K(+) current and Na(+) current*. Eur J Pharmacol, 2019. **844**: p. 95-101.

Reviewers' Comments:

Reviewer #1:

Remarks to the Author:

The authors have addressed my concerns.

Reviewer #2:

Remarks to the Author:

There are no further comments/ concerns.